# A conserved filamentous assembly underlies the structure of the meiotic chromosome axis

Alan MV West[1,2†], Scott C Rosenberg[3†‡], Sarah N Ur[1], Madison K Lehmer[2§], Qiaozhen Ye[2], Götz Hagemann[4], Iracema Caballero[5], Isabel Usón[5,6], Amy J MacQueen[7], Franz Herzog[4], Kevin D Corbett[2,3,8]*

[1]Biomedical Sciences Graduate Program, University of California, San Diego, La Jolla, United States; [2]Department of Cellular and Molecular Medicine, University of California, San Diego, La Jolla, United States; [3]Department of Chemistry, University of California, San Diego, La Jolla, United States; [4]Gene Center and Department of Biochemistry, Ludwig-Maximilians-Universität München, Munich, Germany; [5]Crystallographic Methods, Institute of Molecular Biology of Barcelona (IBMB-CSIC), Barcelona, Spain; [6]Institució Catalana de Recerca i Estudis Avançats (ICREA), Barcelona, Spain; [7]Department of Molecular Biology and Biochemistry, Wesleyan University, Middletown, United States; [8]Ludwig Institute for Cancer Research, La Jolla, United States

*For correspondence:
kcorbett@ucsd.edu

[†]These authors contributed equally to this work

Present address: [‡]Genentech Inc, South San Francisco, United States; [§]Molecular & Cell Biology Graduate Program, University of California, Berkeley, Berkeley, United States

Competing interests: The authors declare that no competing interests exist.

**Abstract** The meiotic chromosome axis plays key roles in meiotic chromosome organization and recombination, yet the underlying protein components of this structure are highly diverged. Here, we show that 'axis core proteins' from budding yeast (Red1), mammals (SYCP2/SYCP3), and plants (ASY3/ASY4) are evolutionarily related and play equivalent roles in chromosome axis assembly. We first identify 'closure motifs' in each complex that recruit meiotic HORMADs, the master regulators of meiotic recombination. We next find that axis core proteins form homotetrameric (Red1) or heterotetrameric (SYCP2:SYCP3 and ASY3:ASY4) coiled-coil assemblies that further oligomerize into micron-length filaments. Thus, the meiotic chromosome axis core in fungi, mammals, and plants shares a common molecular architecture, and likely also plays conserved roles in meiotic chromosome axis assembly and recombination control.
DOI: https://doi.org/10.7554/eLife.40372.001

## Introduction

Meiosis is a specialized cell division program that generates haploid gametes from a diploid cell, in preparation for sexual reproduction. Meiosis achieves a two-fold reduction in ploidy through two successive cell divisions without an intervening DNA replication step. Homologous chromosomes segregate from one another in the first meiotic division (meiosis I), and replicated sister chromosomes segregate in meiosis II. Accurate segregation of homologs in meiosis I requires that homologs identify and physically link to one another in the extended meiotic prophase. Homolog recognition and physical association is achieved through crossover formation, in which programmed double strand DNA breaks (DSBs) in each chromosome are repaired in a specialized homologous recombination pathway, resulting in a reciprocal exchange of genetic information and the physical linkage of homologs.

A highly-conserved meiosis-specific structure, the chromosome axis, assembles in early meiotic prophase and provides a scaffold for the organization of chromosomes as a linear array of loops

(*van Heemst and Heyting, 2000*; *Zickler and Kleckner, 1999*), and also orchestrates the formation of DSBs and their repair as inter-homolog crossovers (*Carballo et al., 2008*; *Hollingsworth, 2010*; *Humphryes and Hochwagen, 2014*; *Kim et al., 2010*; *Lao and Hunter, 2010*; *Lao et al., 2013*; *Niu et al., 2007*; *Niu et al., 2009*; *Panizza et al., 2011*; *Subramanian and Hochwagen, 2014*; *Zickler and Kleckner, 2015*). Components of the chromosome axis include DNA-binding and -organizing cohesin complexes (*Onn et al., 2008*), plus proteins of the meiotic HORMAD family that regulate DSB and crossover formation (*Aravind and Koonin, 1998*; *Caryl et al., 2000*; *Couteau and Zetka, 2005*; *Hollingsworth and Byers, 1989*; *Lorenz et al., 2004*; *Martinez-Perez and Villeneuve, 2005*; *Vader and Musacchio, 2014*; *Wojtasz et al., 2009*; *Zetka et al., 1999*). These proteins are named after their N-terminal HORMA (Hop1, Rev7, Mad2) domain, a conserved fold that interacts with short sequence motifs termed 'closure motifs' (*Aravind and Koonin, 1998*; *Rosenberg and Corbett, 2015*). We have previously shown that meiotic HORMADs in *C. elegans* (HIM-3, HTP-1, HTP-2, and HTP-3) form a large oligomeric complex on the chromosome axis through interactions between their N-terminal HORMA domains and closure motifs in their C-termini (*Kim et al., 2014*). We later showed that meiotic HORMAD proteins from both mammals (HORMAD1 and HORMAD2) and *S. cerevisiae* (Hop1) also possess putative closure motifs at their C-termini (*Kim et al., 2014*; *West et al., 2018*), suggesting that head-to-tail self-assembly of these proteins is conserved and important for their DSB- and crossover-promoting functions.

In addition to HORMAD proteins, most organisms also possess additional factors, here termed 'axis core' proteins after Moses (*Moses, 1956*), that are important for axis formation and meiotic HORMAD recruitment. The archetypal axis core protein is *S. cerevisiae* Red1, an 827-residue protein that recruits the HORMAD protein Hop1 to the axis via a putative closure motif in its central region (*West et al., 2018*; *Woltering et al., 2000*). A conserved region at the Red1 C-terminus is predicted to adopt a coiled-coil structure and mediates self-association of the protein (*Hollingsworth and Ponte, 1997*; *Woltering et al., 2000*), suggesting that oligomer formation by Red1 may also be important for axis function. While clearly-identifiable Red1 homologs do not exist outside fungi, many other organisms possess functionally-equivalent axis core proteins with predicted coiled-coil structure. Mammals possess two such proteins, SYCP2 (1500 residues in *Mus musculus*) and SYCP3 (254 residues), which are both required for proper axis formation and wild-type levels of crossovers (*Yuan et al., 2000*; *Yuan et al., 2002*), and are known to interact with one another through their C-terminal coiled-coil domains (*Yang et al., 2006*). SYCP2 and SYCP3 are interdependent for their axis localization (*Pelttari et al., 2001*; *Shin et al., 2010*; *Yang et al., 2006*; *Yuan et al., 2000*), and a mutant of SYCP2 lacking its C-terminal predicted coiled-coil region shows a loss of SYCP3 from the axis (*Shin et al., 2010*; *Yang et al., 2006*). These data suggest that the SYCP2 N-terminal region mediates localization of the complex, while the C-terminal domain mediates oligomerization with SYCP3. In plants, the axis proteins ASY3 (793 residues in *Arabidopsis thaliana*) and ASY4 (212 residues) are both important for crossover formation, and associate with one another through their C-terminal predicted coiled coil domains (*Chambon et al., 2018*; *Ferdous et al., 2012*; *Osman et al., 2018*). While neither SYCP2/SYCP3 nor ASY3/ASY4 have been reported to possess HORMAD-interacting closure motifs, the similar domain structure and roles in crossover formation between these proteins and *S. cerevisiae* Red1 suggests that they may be evolutionarily related (*Ferdous et al., 2012*; *Offenberg et al., 1998*).

In addition to its roles in chromosome organization and crossover formation in early meiotic prophase, the chromosome axis plays a later role as a key structural element of the highly-conserved yet functionally enigmatic synaptonemal complex (SC). As inter-homolog crossovers form, the chromosome axes of each homolog pair, now termed 'lateral elements' of the SC, become linked by coiled-coil 'transverse filaments' along their entire length (*Page and Hawley, 2004*; *Rockmill et al., 1995*; *Sym et al., 1993*). In fungi, plants, and mammals, SC assembly is tightly coordinated with removal of the meiotic HORMADs from the chromosome axis by the AAA+-family ATPase Pch2/TRIP13, in a key feedback mechanism controlling crossover levels (*Börner et al., 2008*; *Joshi et al., 2009*; *Lambing et al., 2015*; *San-Segundo and Roeder, 1999*; *Vader, 2015*). SC assembly is required for crossover maturation, and serves as a signal to the cell that a given homolog pair has obtained crossovers (*Page and Hawley, 2004*; *Zickler and Kleckner, 1999*).

While the overall architecture of the SC—including the lateral elements, transverse filaments, and central element—are becoming better understood (*Cahoon et al., 2017*; *Davies et al., 2012*; *Dunce et al., 2018*; *Köhler et al., 2017*; *Lu et al., 2015*; *Schücker et al., 2015*), the molecular

interactions underlying the chromosome axis, and whether these interactions are conserved across eukaryotes, remain less well-characterized. Specifically, it is not known whether mammalian and plant axis core proteins possess HORMAD-binding closure motifs like Red1, leaving open the question of how HORMADs are recruited to chromosomes in these organisms. More significantly, the oligomeric structure of the axis core proteins, whether this structure is conserved, and how this structure contributes to the axis's roles in chromosome organization, inter-homolog recombination, and SC architecture are important open questions. Mammalian SYCP3 is known to form coiled-coil homotetramers (*Syrjänen et al., 2014*) that self-associate into larger structures both in cell culture (*Pelttari et al., 2001*) and in vitro (*Syrjänen et al., 2014*), but how SYCP3 cooperates with SYCP2 to mediate chromosome localization and axis assembly is not known. Neither fungal Red1 nor plant ASY3/ASY4 have been characterized biochemically, leaving open the question of how these proteins self-assemble, and whether these assemblies resemble those of mammalian SYCP3.

Here, we address these questions and establish that the molecular architecture of the meiotic chromosome axis is shared between fungi, mammals and plants. We find that budding-yeast Red1 forms stable homotetrameric complexes via its coiled-coil C-terminus, and that these tetramers associate end-to-end to form extended filaments visible by electron microscopy. We identify HORMAD-binding closure motifs in both mammalian SYCP2 and plant ASY3, supporting these proteins' identification as Red1 homologs and strongly suggesting a role in meiotic HORMAD recruitment to meiotic chromosomes. We further show that both SYCP2/SYCP3 and ASY3/ASY4 form heterotetrameric coiled-coil complexes that self-assemble into extended filaments, paralleling our findings with Red1. Taken together, these data reveal common principles of meiotic chromosome axis assembly and function that are widely shared throughout eukaryotes.

## Results

### Budding Red1 forms filaments from coiled-coil tetramer units

In budding yeast, the chromosome axis is made up of the HORMAD protein Hop1, its binding partner Red1, and cohesin complexes containing the meiosis-specific kleisin subunit Rec8 (*Klein et al., 1999*; *Zickler and Kleckner, 1999*). We and others have outlined the assembly mechanisms of Hop1, which binds short 'closure motifs' in its own C-terminal tail and in Red1 through its conserved HORMA domain (*Figure 1A*) (*West et al., 2018*; *Woltering et al., 2000*). Red1 is less well-understood. This protein possesses a conserved N-terminal domain immediately followed by a Hop1-binding closure motif, an extended linker domain with high predicted disorder, and a C-terminal domain that mediates Red1 self-association and is predicted to adopt a coiled-coil structure (*Figure 1A*) (*Hollingsworth and Ponte, 1997*; *West et al., 2018*; *Woltering et al., 2000*). Prior genetic studies isolated two point-mutations in the Red1 C-terminal domain, I743A (*Eichinger and Jentsch, 2010*) and I758R (*Lin et al., 2010*), that each strongly affect both SC assembly and spore viability in *S. cerevisiae*. While these phenotypes were attributed to effects on binding other meiotic chromosome-associated proteins, these residues' location within a predicted coiled-coil domain prompted us to consider instead that the observed defects may be due to disruption of a Red1 oligomer important for meiotic chromosome axis function.

To test this idea, we expressed in *E. coli*, and purified the Red1 C-terminal domain from several budding yeasts, and found that uniformly, these proteins formed large assemblies as measured by size-exclusion chromatography (*Figure 1B* and data not shown). We examined one Red1 construct, *Zygosaccharomyces rouxii* (*Zr*) Red1$^{705-798}$, by negative-stain electron microscopy. We observed filaments up to several microns in length (*Figure 1C*), suggesting that the large assemblies of purified Red1 C-terminal domain are not disordered aggregates but rather represent a biologically relevant structure. In the course of construct optimization, we also cloned and purified a truncated *Zr* Red1 construct missing the C-terminal seven residues of the protein (*Zr* Red1$^{705-791}$). Strikingly, this construct did not form assemblies in solution, but rather formed stable homotetramers as measured by size-exclusion chromatography coupled to multi-angle light scattering (SEC-MALS; *Figure 1D–F*). Together, these data suggest that the Red1 C-terminal domain forms coiled-coil homotetramers that associate end-to-end to form extended filaments.

We next examined the effects of mutating *Zr* Red1 I715 and M730, which are equivalent to *Sc* Red1 I743 and I758, respectively. We mutated both residues to arginine and examined the effects

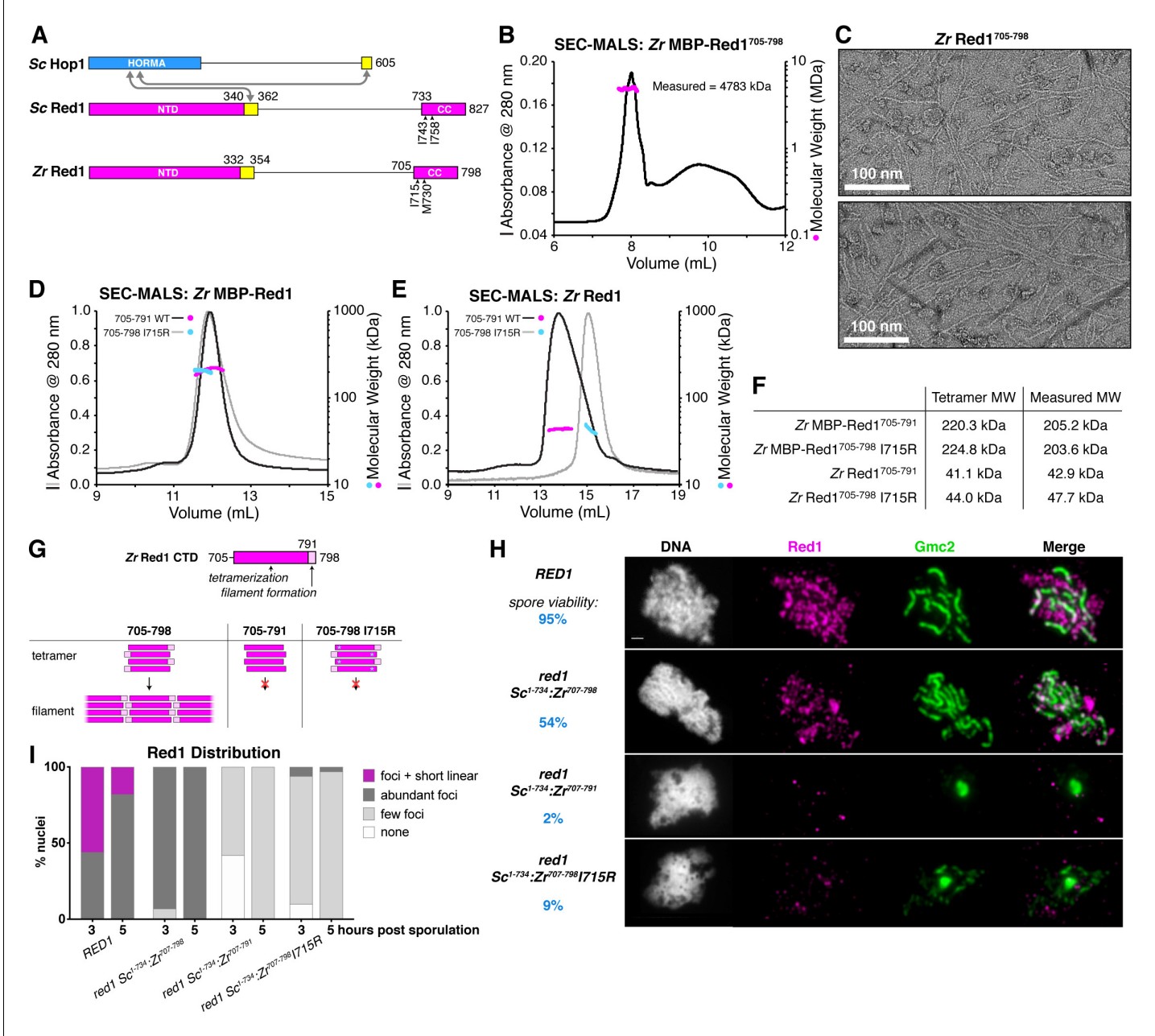

**Figure 1.** Molecular architecture of the budding-yeast Red1 C-terminal domain. (**A**) Schematic of *S. cerevisiae* (*Sc*) chromosome axis proteins Hop1 and Red1, and *Zygosaccharocmyces zouxii* (*Zr*) Red1. Yellow regions indicate Hop1-binding closure motifs (***West et al., 2018***). For *S. cerevisiae* Red1, the positions of two previously-identified mutations in the C-terminal domain that disrupt axis function, I743A (***Eichinger and Jentsch, 2010***) and I758R (***Lin et al., 2010***), are shown. See **Figure 1—figure supplement 1A** for sequence alignment of the Red1 C-terminal domain. (**B**) SEC-MALS analysis of purified His$_6$-MBP-*Zr* Red1$^{705-798}$. Calculated molecular weight of a monomer = 55.9 kDa; Measured molecular weight = 4783 kDa (~85 mer). (**C**) Representative negative-stain electron micrographs of purified untagged *Zr* Red1$^{705-798}$. See **Figure 1—figure supplement 2A** for additional full micrographs, and **Figure 1—figure supplement 2B** for micrographs of His$_6$-MBP-*Zr* Red1$^{705-798}$. (**D**) SEC-MALS analysis of purified His$_6$-MBP-*Zr* Red1$^{705-791}$ and His$_6$-MBP-*Zr* Red1$^{705-798}$ I715R. (**E**) SEC-MALS analysis of purified *Zr* Red1$^{705-791}$ and *Zr* Red1$^{705-798}$ I715R (as in panel D, but with His$_6$-MBP tag removed). (**F**) Table summarizing SEC-MALS results from (**D**) and (**E**). (**G**) Schematic of *Zr* Red1 C-terminal domain oligomerization. Wild-type *Zr* Red1$^{705-798}$ forms homotetramers that further oligomerize into extended filaments. Removal of the C-terminal seven amino acids (*Zr* Red1$^{705-791}$) or mutations of I715 to arginine (*Zr* Red1$^{705-798}$ I715R) results in loss of filament formation but maintenance of tetramer formation. (**H**) Representative surface-spread mid-meiotic prophase nuclei from wild-type (top row), and *red1* mutant alleles: *red1-Sc*$^{1-734}$:*Zr*$^{707-798}$ (second row), *red1-Sc*$^{1-734}$:*Zr*$^{707-791}$ (third row), and *red1-Sc*$^{1-734}$:*Zr*$^{707-798}$ I715R] (bottom row). Spore viability for each homozygous strain is shown in blue (n = 52–128, see Materials and Methods). Mid-meiotic prophase chromosomes are stained with DAPI to label DNA (white), anti-Red1 (magenta), and anti-Gmc2 (green). Scale bar, 1 μm. See **Figure 1—figure supplement 3B–E** for additional images. (**I**) Quantification of the distribution of Red1 on meiotic chromosomes

*Figure 1 continued on next page*

**eLIFE** Research article

Chromosomes and Gene Expression | Structural Biology and Molecular Biophysics

*Figure 1 continued*

at 3 hr and 5 hr after introduction into sporulation medium (n = 30–50 cells for each strain and time-point). 'foci + short linear'=cells with abundant foci and short linear stretches of Red1 staining; 'abundant foci'=cells with more than 25 strong Red1 foci; 'few foci'=cells with fewer than 25 weak Red1 foci. See *Figure 1—figure supplement 3F–H* for further quantification of Red1, Gmc2, and polycomplex assembly.

DOI: https://doi.org/10.7554/eLife.40372.002

The following figure supplements are available for figure 1:

**Figure supplement 1.** Sequence and structural analysis of the fungal Red1 C-terminus.

DOI: https://doi.org/10.7554/eLife.40372.003

**Figure supplement 2.** Electron microscopy of the fungal Red1 C-terminus.

DOI: https://doi.org/10.7554/eLife.40372.004

**Figure supplement 3.** Red1 chromosome localization and synaptonemal complex assembly in chimeric *Sc-Zr red1* strains.

DOI: https://doi.org/10.7554/eLife.40372.005

by SEC-MALS. We found that the *Zr* Red1$^{705-798}$ I715R mutant formed a homotetramer in solution, instead of the extended filaments formed by wild-type *Zr* Red1$^{705-798}$ (*Figure 1D–F*). The *Zr* Red1$^{705-798}$ M730R mutant was poorly behaved in solution, precluding a detailed analysis of this mutant's effects on filament formation. We next examined the effect of mutating I743 and I758 in *S. cerevisiae* Red1, which was soluble in vitro only when fused to a maltose binding protein (MBP) solubility tag. Using this system, we found that the wild-type *Sc* Red1 C-terminal domain (residues 731–827) forms large assemblies (*Figure 1—figure supplement 1B*). Mutating I743 to arginine disrupted higher-order assembly but maintained tetramer formation, and mutating I758 to arginine completely disrupted Red1 self-assembly resulting in monomers (*Figure 1—figure supplement 1B*). We were unable to determine the effect of truncating the *Sc* Red1 C-terminus due to poor expression.

When combined with prior findings that the *Sc RED1*-I743A mutant shows low spore viability, our finding that mutating *Sc* Red1 I743 specifically disrupts filament assembly suggests that filament formation by Red1 may be critical for meiotic chromosome axis structure and function. To test this idea, we replaced the coiled-coil region of *S. cerevisiae* Red1 (residues 734–827) with the equivalent region of *Zr* Red1 (residues 707–798) to generate a chimeric Red1 protein (*red1-Sc$^{1-734}$:Zr$^{707-798}$*) that we could engineer with predictable effects based on our in vitro data. We next specifically disrupted filament formation in this chimeric construct by removing residues 792–798 (*red1-Sc$^{1-734}$:Zr$^{707-791}$*) or mutating *Zr* Red1 I715 to arginine (*red1-Sc$^{1-734}$:Zr$^{707-798}$I715R*). All three chimeric Red1 constructs were expressed equivalently to wild-type Red1 (*Figure 1—figure supplement 3A*), but only the full-length chimera supported appreciable levels of spore viability (54% viable spores for *red1-Sc$^{1-734}$:Zr$^{707-798}$* versus 2% for *red1-Sc$^{1-734}$:Zr$^{707-791}$* and 9% for *red1-Sc$^{1-734}$:Zr$^{707-798}$I715R*; *Figure 1H*). We next examined chromosome localization of the chimeric Red1 constructs in meiotic prophase, and their ability to support synaptonemal complex assembly. We found that the full-length chimeric protein (Red1-*Sc$^{1-734}$:Zr$^{707-798}$*) localized robustly to meiotic chromosomes and supported synaptonemal complex assembly, albeit less efficiently than wild-type Red1 (*Figure 1H–I*). Both truncation of the Red1 C-terminus (Red1-*Sc$^{1-734}$:Zr$^{707-791}$*) and the I715R mutation (Red1-*Sc$^{1-734}$:Zr$^{707-798}$I715R*) caused a strong defect in chromosome localization of Red1, and a near-complete loss of synaptonemal complex formation with a corresponding increase in polycomplex formation (*Figure 1H–I*, *Figure 1—figure supplement 3*). In both mutant strains, chromosomes were also less well-defined in DAPI staining than in either wild-type or Red1-*Sc$^{1-734}$:Zr$^{707-798}$* cells (*Figure 1H*, *Figure 1—figure supplement 3*), suggesting defects in chromosome axis assembly and chromosome compaction. We conclude that Red1 filament formation is important for robust chromosome localization of Red1, and absolutely critical for proper assembly of the chromosome axis and, by extension, the synaptonemal complex. Finally, these data also suggest that the previously-identified deleterious effects of the *Sc* Red1 I743A and I758R mutations (*Eichinger and Jentsch, 2010*; *Lin et al., 2010*) may be due to disruption of Red1 filament assembly.

## SYCP2 is an interaction hub for the mammalian chromosome axis

The mammalian chromosome axis comprises cohesin complexes with several meiosis-specific subunits (*Biswas et al., 2016*; *Fukuda et al., 2014*; *Ward et al., 2016*; *Winters et al., 2014*); two meiotic HORMAD proteins, HORMAD1 and HORMAD2 (*Fukuda et al., 2010*); and the coiled-coil proteins SYCP2 and SYCP3 (*Li et al., 2011*; *Llano et al., 2012*). We have previously shown that both

HORMAD1 and HORMAD2 possess short motifs at their extreme C-termini that associate with these proteins' N-terminal HORMA domains (*Kim et al., 2014*), strongly suggesting that these motifs constitute closure motifs equivalent to that previously identified in the *S. cerevisiae* Hop1 C-terminus (*Niu et al., 2005*; *West et al., 2018*). SYCP2 has been proposed as a distant homolog of budding-yeast Red1, and possesses a similar domain structure: an N-terminal ordered domain that may mediate the protein's association with chromosomes (*Feng et al., 2017*), followed by an extended disordered region and a C-terminal domain of ~175 residues predicted to form a coiled-coil. Instead of self-associating like Red1, however, the SYCP2 coiled-coil domain binds the shorter coiled-coil protein SYCP3 (*Figure 2A*, *Figure 2—figure supplement 1*) (*Tarsounas et al., 1997*; *Yang et al., 2006*). Additionally, co-expression of SYCP2 and SYCP3 in cultured cells results in the assembly of large filamentous structures that incorporate both proteins, suggesting a capacity for self-assembly of SYCP2:SYCP3 complexes (*Pelttari et al., 2001*).

To outline protein-protein interactions within the mammalian chromosome axis, we used yeast two-hybrid assays to test for interactions between SYCP2, SYCP3, and HORMAD2. We identified a short region of SYCP2 directly following the protein's ordered N-terminal domain (residues 395–414) that binds the HORMAD2 HORMA domain in both yeast two-hybrid and when co-expressed in *E. coli* (*Figure 2B*, *Figure 2—figure supplement 2A–C*). This region shares homology to HORMAD1 and HORMAD2 C-termini, suggesting that it constitutes a closure motif (*Figure 2A*, *Figure 2—figure supplement 3*). The location of the putative SYCP2 closure motif—directly following the ordered N-terminal domain—is also equivalent to the location of the budding-yeast Red1 closure motif, lending support to the idea that SYCP2 and Red1 are homologs. We directly tested binding of the isolated HORMAD2 HORMA domain (residues 1–241) to a peptide encoding the putative closure motif of HORMAD2, and detected robust binding (*Figure 2—figure supplement 2D*). Further, a pre-assembled complex of HORMAD2 and the putative SYCP2 closure motif showed no binding to the HORMAD2 closure motif peptide, indicating that these sequences compete for binding to the HORMAD2 HORMA domain (*Figure 2—figure supplement 2D*). Finally, despite the overall similarity between HORMAD1 and HORMAD2, we have so far been unable to demonstrate an interaction between SYCP2 and HORMAD1. While this is at least partially due to poor expression and solubility of *M. musculus* HORMAD1 in our assays (not shown), it remains possible that SYCP2 only interacts directly with HORMAD2, while HORMAD1 is recruited by HORMAD2 (*Kim et al., 2014*) and potentially other chromosome axis components.

Our yeast two-hybrid assays also confirmed that the coiled-coil regions of SYCP2 and SYCP3 associate (*Figure 2B*). We next sought to purify an SYCP2:SYCP3 complex, in order to characterize its structure and oligomeric state. We co-expressed the coiled-coil domains of *M. musculus* SYCP2 (residues 1325–1500) and SYCP3 (residues 84–248), and found that the proteins form a stoichiometric complex (*Figure 2C*) that, like the Red1 C-terminal domain, forms large assemblies in vitro as judged by size-exclusion chromatography (*Figure 3A,B*). Analysis of *Mm* SYCP2$^{1325-1500}$:SYCP3$^{84-248}$ assemblies by negative-stain electron microscopy revealed extended filaments much like those we observed for *Zr* Red1$^{705-798}$ (*Figure 2D*, *Figure 2—figure supplement 4A*). When we visualized the same complex with SYCP2 tagged at its N-terminus with MBP (*Mm* MBP-SYCP2$^{1325-1500}$:SYCP3$^{84-248}$) we observed filaments decorated with regularly-spaced pairs of densities ~5 nm in diameter, equivalent to the expected size of a single ~43 kDa MBP monomer (*Figure 2E*, *Figure 2—figure supplement 4B*). We measured the inter-MBP spacing along SYCP2:SYCP3 filaments, and found an average spacing of 23.1 nm, which closely matches the expected length of an α-helical coiled coil ~175 residues in length (*Figure 2F*). These data suggest that the SYCP2:SYCP3 complex forms filaments through end-to-end association of individual α-helical units ~23 nm in length, with each unit containing two copies of SYCP2.

The experiments above were conducted with a construct of SYCP3, residues 84–248, lacking the C-terminal six residues of this protein. These residues have been previously shown to be critical for formation of large homotypic SYCP3 filaments when the protein is overexpressed in mammalian tissue-culture cells (*Baier et al., 2007*; *Yuan et al., 1998*), and for formation of large SYCP3 assemblies in vitro (*Syrjänen et al., 2014*).We next purified an SYCP2:SYCP3 complex containing these residues, *Mm* SYCP2$^{1325-1500}$:SYCP3$^{84-254}$, and visualized the complex by negative-stain electron microscopy. We found that this complex forms filaments equivalent to *Mm* SYCP2$^{1325-1500}$:SYCP3$^{84-248}$, but that in contrast to the truncated complex, filaments containing the full SYCP3 C-terminus tended to self-associate into bundles (*Figure 2G*). Given the high conservation of these residues and their

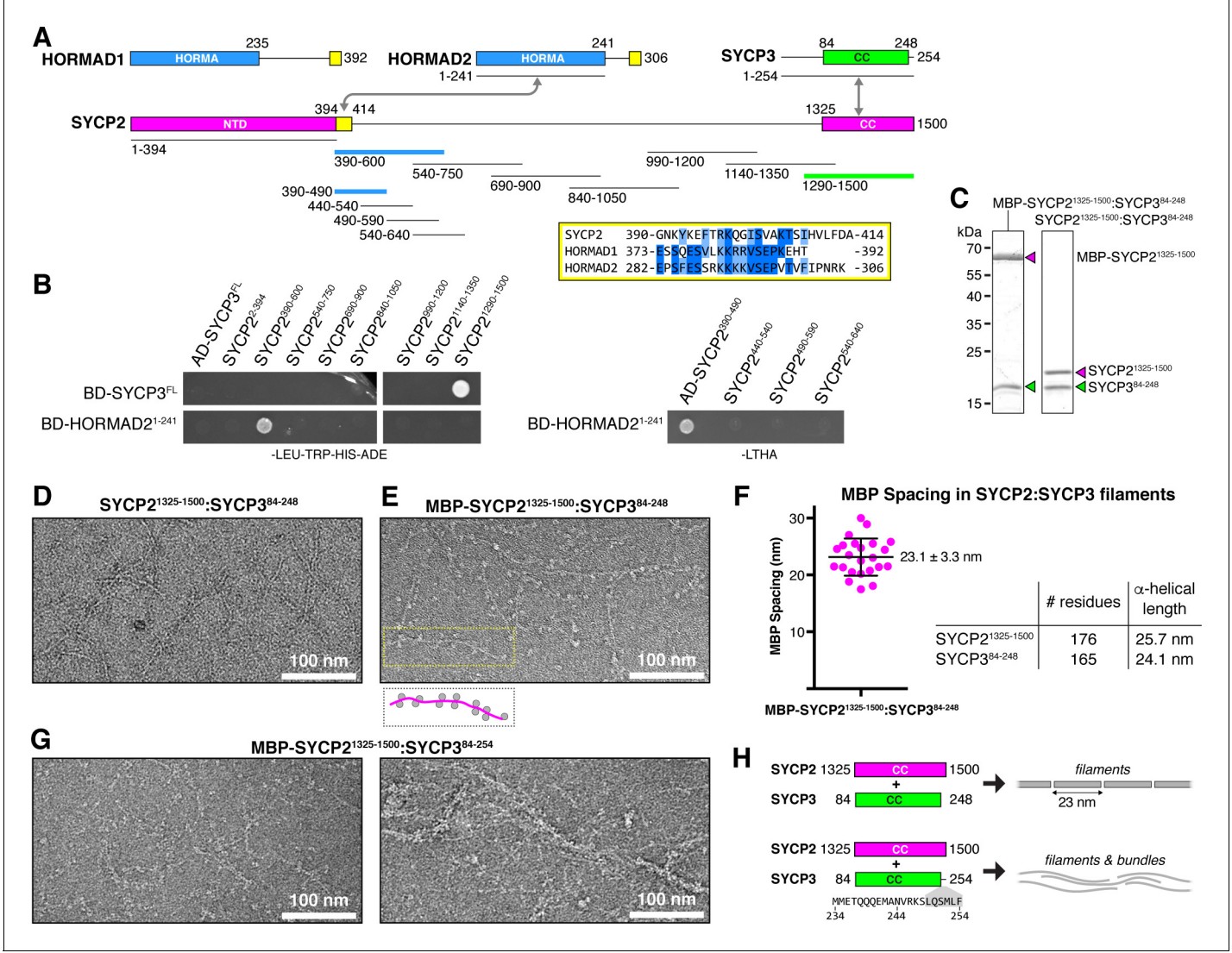

**Figure 2.** Protein-protein interactions and filament formation by mammalian SYCP2 and SYCP3. (A) Schematic of *M. musculus* (*Mm*) chromosome axis proteins, with underlines indicating fragments used for yeast two-hybrid analysis. The SYCP2 NTD (residues 1–394) forms a globular structure with unknown function (*Feng et al., 2017*). See *Figure 2—figure supplement 1* for detailed coiled-coil and alpha-helix predictions of the SYCP2 and SYCP3 C-terminal domains. Yellow regions indicate putative HORMAD-binding closure motifs (*Figure 2—figure supplement 3*). (B) Yeast two-hybrid analysis of SYCP2 truncations versus SYCP3 and the HORMAD2 HORMA domain (residues 1-241). AD: Gal4 activation domain fusion; BD: Gal4 DNA-binding domain fusion. Stringent selection on -LEU-TRP-HIS-ADE (-LTHA) media is shown; see *Figure 2—figure supplement 2* for complete yeast two-hybrid results and for coexpression of SYCP2 fragments with HORMAD2$^{1-241}$. (C) SDS-PAGE analysis of purified *Mm* SYCP2$^{1325-1500}$:SYCP3$^{84-248}$ complexes, with an N-terminal MBP tag on SYCP2 (left) or with the tag removed (right). (D) Representative negative-stain electron micrograph of purified *Mm* SYCP2$^{1325-1500}$:SYCP3$^{84-248}$. See *Figure 2—figure supplement 4A* for additional full micrographs. (E) Representative negative-stain electron micrographs of purified His$_6$-MBP-*Mm* SYCP2$^{1325-1500}$:SYCP3$^{84-248}$. See *Figure 2—figure supplement 4B* for additional full micrographs. (F) Quantification of inter-MBP spacing in micrographs of His$_6$-MBP-*Mm* SYCP2$^{1325-1500}$:SYCP3$^{84-248}$ filaments. The measured spacing of 23.1 ± 3.3 nm (mean ±standard deviation from 23 measured intervals) is equivalent to the length of a ~160 residue coiled-coil (0.146 nm rise per residue). (G) Representative negative-stain electron micrographs of purified His$_6$-MBP-*Mm* SYCP2$^{1325-1500}$:SYCP3$^{84-254}$. See *Figure 2—figure supplement 5* for additional full micrographs. (H) Schematic summary of negative-stain electron microscopy results: *Mm* SYCP2$^{1325-1500}$:SYCP3$^{84-248}$ forms individual filaments assembled from ~23 nm units, while re-addition of the highly-conserved C-terminal six residues of SYCP3 (249–254; shown in gray below schematic) causes self-association/bundling of these filaments.

DOI: https://doi.org/10.7554/eLife.40372.006

The following figure supplements are available for figure 2:

**Figure supplement 1.** Coiled-coil and helical predictions for mammalian SYCP2 and SYCP3.

DOI: https://doi.org/10.7554/eLife.40372.007

**Figure supplement 2.** Protein-protein interactions in the mammalian meiotic chromosome axis.

*Figure 2 continued on next page*

*Figure 2 continued*

DOI: https://doi.org/10.7554/eLife.40372.008

**Figure supplement 3.** Alignment of putative closure motifs in mammalian SYCP2, HORMAD1, and HORMAD2.

DOI: https://doi.org/10.7554/eLife.40372.009

**Figure supplement 4.** Electron micrographs of *Mm* SYCP2$^{1325-1500}$:SYCP3$^{84-248}$ filaments.

DOI: https://doi.org/10.7554/eLife.40372.010

**Figure supplement 5.** Electron micrographs of *Mm* SYCP2$^{1325-1500}$:SYCP3$^{84-254}$ filaments and bundles.

DOI: https://doi.org/10.7554/eLife.40372.011

importance for large-scale SYCP3 assembly in multiple assays, we propose that the SYCP3 C-terminus may mediate bundling of SYCP2:SYCP3 filaments as an important step in assembly of the mammalian meiotic chromosome axis (*Figure 2H*). As we do not observe bundling in filaments of budding-yeast Red1 (*Figure 1C*) or plant axis core proteins (see below), this tendency to bundle may be specific to the mammalian chromosome axis.

We next sought to further dissect the SYCP2:SYCP3 filament assembly. We progressively truncated both proteins, and found that removal of 21 residues from either the SYCP2 C-terminus (residues 1480–1500) or the SYCP3 N-terminus (residues 84–104) led to a strong reduction in filament formation, and the appearance of a well-defined smaller complex, as measured by size-exclusion chromatography (*Figure 3A,B*). Importantly, these truncations reduced filament assembly while maintaining the association between the two proteins (*Figure 3B*). We combined the truncations on both proteins to yield a minimal construct, *Mm* SYCP2$^{1325-1479}$:SYCP3$^{105-248}$, which we term SYCP2$^{CC}$:SYCP3$^{CC}$ hereon. We first measured the molecular weight of the SYCP2$^{CC}$:SYCP3$^{CC}$ complex with an N-terminal MBP tag on SYCP2$^{CC}$ by SEC-MALS. The measured molecular weight of this

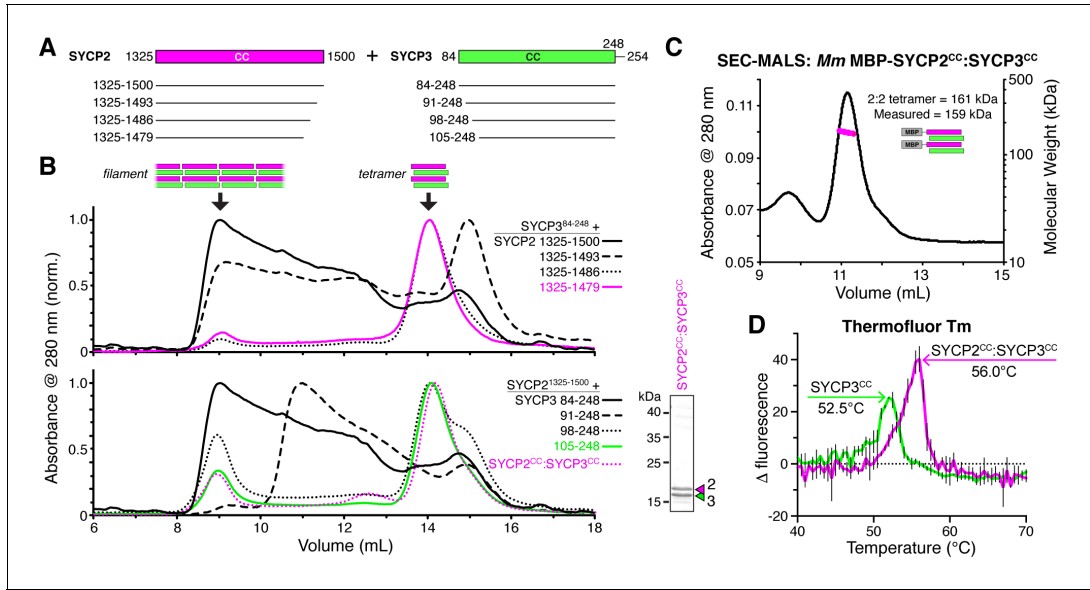

**Figure 3.** SYCP2:SYCP3 filaments are assembled from end-to-end interactions of a 2:2 heterotetrameric unit. (A) Schematic of the predicted coiled-coil regions of *M. musculus* SYCP2 and SYCP3, with truncations used for co-expression/size exclusion chromatography analysis in panels (B) and (C). (B) Superose-6 size exclusion chromatography analysis of truncated *Mm* SYCP2:SYCP3 complexes. All complexes were purified after co-expression using an N-terminal His$_6$-MBP tag on SYCP2. *Upper panel:* truncation of the SYCP2 coiled-coil C-terminus, from 1325 to 1500 (black) to 1325–1479 (magenta), all co-expressed with SYCP3$^{84-248}$. *Lower panel:* truncation of SYCP3 coiled-coil N-terminus, from 84 to 248 (black) to 105–248 (green), all co-expressed with SYCP3$^{1325-1500}$. Magenta dotted line indicates elution profile of MBP-SYCP2$^{1325-1479}$:SYCP3$^{105-248}$ complex (*Mm* SYCP2$^{CC}$:SYCP3$^{CC}$), used for SEC-MALS in panel (C). *Lower right:* SDS-PAGE analysis of purified *Mm* SYCP2$^{CC}$:SYCP3$^{CC}$ complex (with His$_6$-MBP tag removed). (C) SEC-MALS analysis of purified His$_6$-MBP-*Mm* SYCP2$^{CC}$:SYCP3$^{CC}$ complex. Calculated molecular weight of a 2:2 heterotetramer = 161.4 kDa; Measured molecular weight = 158.6 kDa. (D) Thermofluor melting-temperature (T$_m$) analysis for *Mm* SYCP2$^{CC}$:SYCP3$^{CC}$ (red) versus a homotetrameric *Mm* SYCP3$^{CC}$ complex (green). Thick colored lines represent an average of three independent measurements, with standard deviation represented by thin vertical black lines.

DOI: https://doi.org/10.7554/eLife.40372.012

complex, 159 kDa, closely matched the predicted molecular weight of 161 kDa for a 2:2 heterotetramer of SYCP2 and SYCP3 (*Figure 3C*).

Prior work on *H. sapiens* SYCP3 has shown that this protein self-associates to form coiled-coil homoetramers in vitro (*Syrjänen et al., 2014*). We found that *Mm* SYCP3$^{CC}$ also forms homoetramers in the absence of SYCP2$^{CC}$ (not shown), and when we determined the structure of *Mm* SYCP3$^{CC}$ by x-ray crystallography, we observed an antiparallel coiled-coil homotetramer similar in structure to *H. sapiens* SYCP3 (*Figure 4—figure supplement 1*). We were unable to determine a structure of the SYCP2$^{CC}$:SYCP3$^{CC}$ heterotetramer. As SYCP2 and SYCP3 share limited sequence homology in their coiled-coil region, we reasoned that SYCP3 homotetramers may form through promiscuous coiled-coil interactions in the absence of SYCP2. To compare the stability of *Mm* SYCP3$^{CC}$ homotetramers with *Mm* SYCP2$^{CC}$:SYCP3$^{CC}$ heterotetramers, we measured their melting temperatures ($T_m$) using a dye-binding assay. We found that the SYCP2$^{CC}$:SYCP3$^{CC}$ heterotetramer is more stable than SYCP3$^{CC}$ on its own (56.0°C $T_m$ versus 52.5°C; *Figure 3D*), supporting the idea that the heterotetrameric complex is the preferred state when both proteins are present.

## The SYCP2:SYCP3 complex is an antiparallel heterotetramer

While the SYCP3$^{CC}$ homotetramer is likely not the favored state in the presence of SYCP2, its structure may nonetheless be informative as to the structure of SYCP2$^{CC}$:SYCP3$^{CC}$. Given its 2:2 stoichiometry and our observed effects on filament formation from truncating opposite ends of SYCP2 and SYCP3, we reasoned that SYCP2$^{CC}$:SYCP3$^{CC}$ may form a complex with two SYCP2 protomers oriented parallel to one another, and antiparallel to two SYCP3 protomers. To test this idea, we generated a series of *Hs* and *Mm* SYCP2:SYCP3 constructs with the two proteins fused end-to-end through a short peptide linker (*Figure 4A*). One such construct, *Hs* SYCP3$^{87-230}$-[GSGASG]-SYCP2$^{1352-1508}$ (termed *Hs* SYCP3$^{CC}$-SYCP2$^{CC}$ fusion hereon), was highly-expressed in *E. coli* and formed a stable dimer by SEC-MALS, equivalent to an SYCP2:SYCP3 heterotetramer (*Figure 4B*). We were unable to crystallize this complex, so we turned instead turned to small-angle x-ray scattering, which provides low-resolution size and shape information on macromolecular complexes in solution. SAXS can provide a reliable measure of a particle's maximum dimension ($d_{max}$) and radius of gyration ($R_g$), as well as, for cylindrical particles, the cross-sectional radius of gyration ($R_c$) (*Feigin and Svergun, 1987*; *Glatter and Kratky, 1982*). Analysis of the *Hs* SYCP3$^{CC}$-SYCP2$^{CC}$ fusion by SAXS showed that this complex's $d_{max}$, $R_g$, and $R_c$ closely match theoretical values calculated from the crystal structure of the *Hs* SYCP3$^{CC}$ homotetramer (*Figure 4C–E*, *Figure 4—figure supplement 2*). Further, the intra-particle distance distribution function calculated from the SAXS scattering curve also closely matched the profile calculated from the *Hs* SYCP3$^{CC}$ crystal structure (*Figure 4C*). We next performed SAXS on the same *Hs* SYCP3$^{CC}$-SYCP2$^{CC}$ fusion containing a ~ 43 kDa MBP tag fused to its N-terminus (*Figure 4—figure supplement 3*). The measured intra-particle distance distribution of this construct agreed closely to a model containing two MBP monomers at the same end of an SYCP2:SYCP3 tetramer, rather than opposite ends, supporting our model in which the two SYCP3$^{CC}$-SYCP2$^{CC}$ monomers are arranged parallel to one another in the complex. Finally, we also performed SAXS analysis on the heterotetrameric *Mm* SYCP2$^{CC}$:SYCP3$^{CC}$ complex (*Figure 4—figure supplement 4*). This complex partially aggregated in solution, precluding detailed analysis, but showed results broadly consistent with the *Hs* SYCP3$^{CC}$-SYCP2$^{CC}$ fusion. Overall, these data show that the SYCP2$^{CC}$:SYCP3$^{CC}$ complex forms an extended coiled-coil tetramer with an overall structure similar to that of the SYCP3 homotetramer.

Our reconstitution and SAXS analysis of the SYCP3$^{CC}$-SYCP2$^{CC}$ fusion supported a model of the SYCP2:SYCP3 tetramer in which two SYCP2 monomers are arranged parallel to one another, and antiparallel to two SYCP3 monomers. To further confirm this model, we used cross-linking mass spectrometry (XLMS), which identifies pairs of lysine residues whose side-chains are in close proximity in a native complex. We identified 55 cross-links in the *Hs* SYCP3$^{CC}$-SYCP2$^{CC}$ fusion construct: 15 within the SYCP3 region, 13 within SYCP2, and 27 between SYCP3 and SYCP2 (*Supplementary file 2*, *Supplementary file 3*). Of the 27 cross-links identified between SYCP2 and SYCP3, ten were observed at least 8 times in our mass spectrometry experiments. Using sequence alignments and the structures of *H. sapiens* and *M. musculus* SYCP3, we generated physical models for SYCP2:SYCP3 where the monomers are arranged either parallel or antiparallel, and mapped all identified crosslinks onto these models (*Figure 4F*, *Figure 4—figure supplement 5*, *Supplementary file 4*). In agreement with our SAXS data, the crosslinking data strongly support a heterotetramer model with

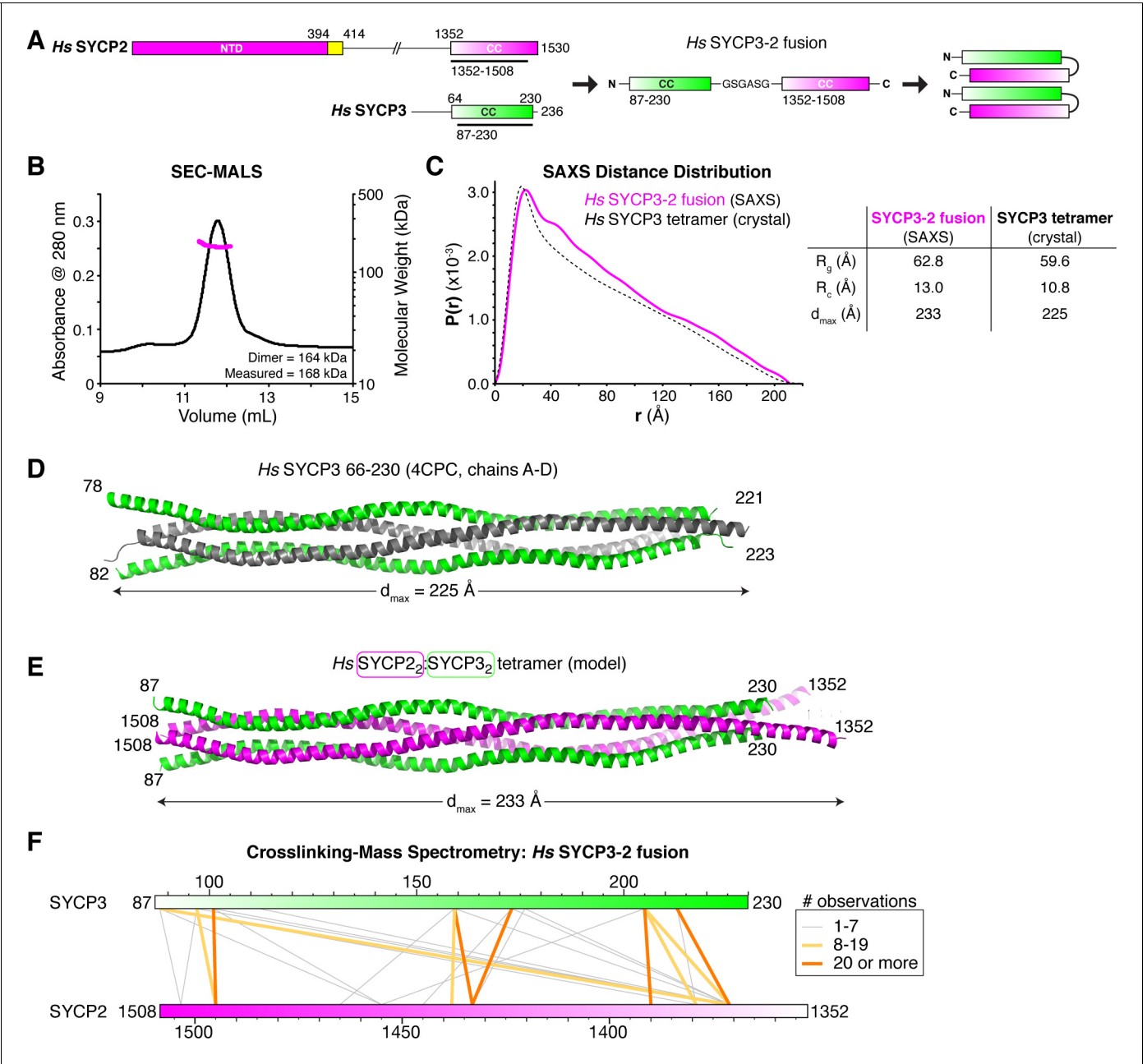

**Figure 4.** Structural analysis of the SYCP2:SYCP3 complex. (**A**) Design of the *H. sapiens* SYCP3$^{CC}$-SYCP2$^{CC}$ fusion, based on the idea that SYCP2 and SYCP3 helices pack antiparallel in the 2:2 heterotetrameric structure. (**B**) SEC-MALS analysis of purified His$_6$-MBP-*Hs* SYCP3$^{CC}$-[GSGASG]-SYCP2$^{CC}$. Measured molecular weight (167.7 kDa) is equivalent to the calculated molecular weight of a homodimer (equivalent to a 2:2 heterotetramer of SYCP2 and SYCP3; 163.7 kDa). (**C**) Intra-particle distance distribution (*P*(*r*)) curve for the *Hs* SYCP3$^{CC}$-SYCP2$^{CC}$ fusion construct derived from small-angle x-ray scattering (SAXS) analysis (magenta), compared to the calculated distance distribution of the *Hs* SYCP3$^{CC}$ homotetramer structure (PDB ID 4CPC; dotted black line) (*Syrjänen et al., 2014*). Lower: table comparing radius of gyration ($R_g$), cross-sectional radius of gyration ($R_c$), and maximum dimensions ($d_{max}$) of the *Hs* SYCP3$^{CC}$-SYCP2$^{CC}$ fusion (calculated from SAXS; see *Figure 4—figure supplement 2*) and the *Hs* SYCP3$^{CC}$ homotetramer (calculated from the crystal structure). See *Figure 4—figure supplement 3* for SAXS analysis of *Hs* MBP-SYCP3$^{CC}$-SYCP2$^{CC}$, and *Figure 4—figure supplement 4* for SAXS analysis of the *Mm* SYCP2$^{CC}$:SYCP3$^{CC}$ complex. (**D**) Structure of the *Hs* SYCP3$^{CC}$ homotetramer structure (PDB ID 4CPC; dotted black line) (*Syrjänen et al., 2014*), with two parallel chains (N-termini left) colored green, and the other two chains (N-termini right) colored gray. We determined the crystal structure of the *M. musculus* SYCP3$^{CC}$ homotetramer in two different crystal forms (*Figure 4—figure supplement 1A–D*). This structure resembles the structure of *Hs* SYCP3$^{CC}$ in the central coiled-coil region, but adopts a distinct, more disordered structure near both ends. (**E**) Model of an *Hs* SYCP2$^{CC}$:SYCP3$^{CC}$ 2:2 heterotetramer, with two SYCP3 chains colored green as in panel (**D**) (N-termini left), and two SYCP2 chains colored magenta (N-termini right). Sequence register was derived from aligning SYCP2 and SYCP3 sequences. (**F**) Schematic of crosslinking mass spectrometry (XLMS) results on the *Hs* SYCP3$^{CC}$-SYCP2$^{CC}$ fusion. Crosslinks observed at least eight times are colored yellow, and crosslinks observed at

*Figure 4 continued on next page*

*Figure 4 continued*

least 20 times are colored orange. See *Supplementary file 2–4* and *Figure 4—figure supplement 5* for full results. 9 of 10 high-scoring crosslinks support the antiparallel subunit arrangement shown in panel (E).

DOI: https://doi.org/10.7554/eLife.40372.013

The following figure supplements are available for figure 4:

**Figure supplement 1.** Structure of *Mm* SYCP3$^{CC}$ homotetramer and modeling of an SYCP2:SYCP3 heterotetramer.
DOI: https://doi.org/10.7554/eLife.40372.014
**Figure supplement 2.** SAXS analysis of *Hs* SYCP3$^{CC}$-SYCP2$^{CC}$ fusion.
DOI: https://doi.org/10.7554/eLife.40372.015
**Figure supplement 3.** SAXS analysis of *Hs* His$_6$-MBP-SYCP3$^{CC}$-SYCP2$^{CC}$ fusion.
DOI: https://doi.org/10.7554/eLife.40372.016
**Figure supplement 4.** SAXS analysis of *Mm* SYCP2$^{CC}$:SYCP3$^{CC}$.
DOI: https://doi.org/10.7554/eLife.40372.017
**Figure supplement 5.** XLMS analysis of *Hs* SYCP3$^{CC}$-SYCP2$^{CC}$ fusion.
DOI: https://doi.org/10.7554/eLife.40372.018

two SYCP2 monomers arranged parallel to one another and antiparallel to two SYCP3 monomers. We propose that these heterotetrameric SYCP2:SYCP3 complexes associate end-to-end to form extended filaments, which can potentially further associate with one another (bundle) through the SYCP3 C-terminus to form the foundation of the chromosome axis.

## Plant ASY3 binds HORMADs and forms filaments with ASY4

In higher plants, the chromosome axis comprises meiosis-specific cohesin complexes (*Bhatt et al., 1999*; *Cai et al., 2003*; *Lam et al., 2005*; *Zamariola et al., 2014*); two meiotic HORMAD proteins, ASY1 and ASY2 (*Caryl et al., 2000*): and two coiled-coil proteins, ASY3 and ASY4 (*Chambon et al., 2018*; *Ferdous et al., 2012*; *Osman et al., 2018*). ASY3 is required for axis localization of ASY1, and its disruption causes a strong defect in crossover formation (*Ferdous et al., 2012*). Despite low sequence identity with either Red1 or SYCP2, ASY3 has been proposed as a functional homolog of Red1 based on phenotypic similarities plus the presence of a conserved C-terminal domain with predicted coiled-coil character (*Figure 5A*) (*Ferdous et al., 2012*). ASY4 was recently identified by two groups as a short protein with high homology to the ASY3 coiled-coil domain, that also interacts with ASY3 (*Chambon et al., 2018*; *Osman et al., 2018*).

To define protein-protein interactions within the plant chromosome axis, we used yeast two-hybrid assays to test interactions between *A. thaliana* ASY1, ASY3, and ASY4. We found that the ASY1 N-terminal HORMA domain (residues 1–234) interacts with its own extreme C-terminus (residues 558–596), revealing that this protein possesses a C-terminal closure motif like its orthologs in *C. elegans*, mammals, and fungi (*Figure 5B*). We further identified an ASY1 HORMA domain-interacting region at the N-terminus of ASY3 (residues 1–50; *Figure 5B*). This region contains a highly-conserved motif of ~30 residues with limited sequence homology to the ASY1 C-terminus (*Figure 5D*, *Figure 5—figure supplement 2*), suggesting that both regions act as HORMAD-binding closure motifs. To verify these interactions, we co-expressed each putative closure motif (fused to an N-terminal His$_6$-MBP tag) with the ASY1 HORMA domain in *E. coli*. Both His$_6$-MBP-ASY3$^{2-50}$ and His$_6$-MBP-ASY1$^{570-596}$ co-purified with untagged ASY1 HORMA domain through Ni$^{2+}$-affinity and size exclusion chromatography (*Figure 5C*), demonstrating a high-affinity interaction. These findings show that plant meiotic HORMADs, like those from fungi and mammals, can interact with closure motif sequences both at their own C-termini and in the N-terminal region of a Red1-like axis core protein.

We next tested interactions between *A. thaliana* ASY3 and ASY4. We found that the C-terminal coiled-coil region of *At* ASY3 (residues 605–793) interacts with ASY4, confirming the recent finding of Osman et al. in *Brassica oleracea* (*Osman et al., 2018*) and of Chambon et al. in *A. thaliana* (*Chambon et al., 2018*) (*Figure 5B*). We next purified a complex between the coiled-coil domains of *A. thaliana* ASY3 and ASY4 (*Figure 5E*), which formed large assemblies in solution as measured by size-exclusion chromatography. Negative-stain electron microscopy on purified His$_6$-MBP-ASY3$^{605-793}$:ASY4$^{FL}$ assemblies revealed extended filaments equivalent to those observed with both budding-yeast Red1 and mammalian SYCP2:SYCP3 (*Figure 5F*). As with the *Mm* MBP-SYCP2$^{1325-1500}$:

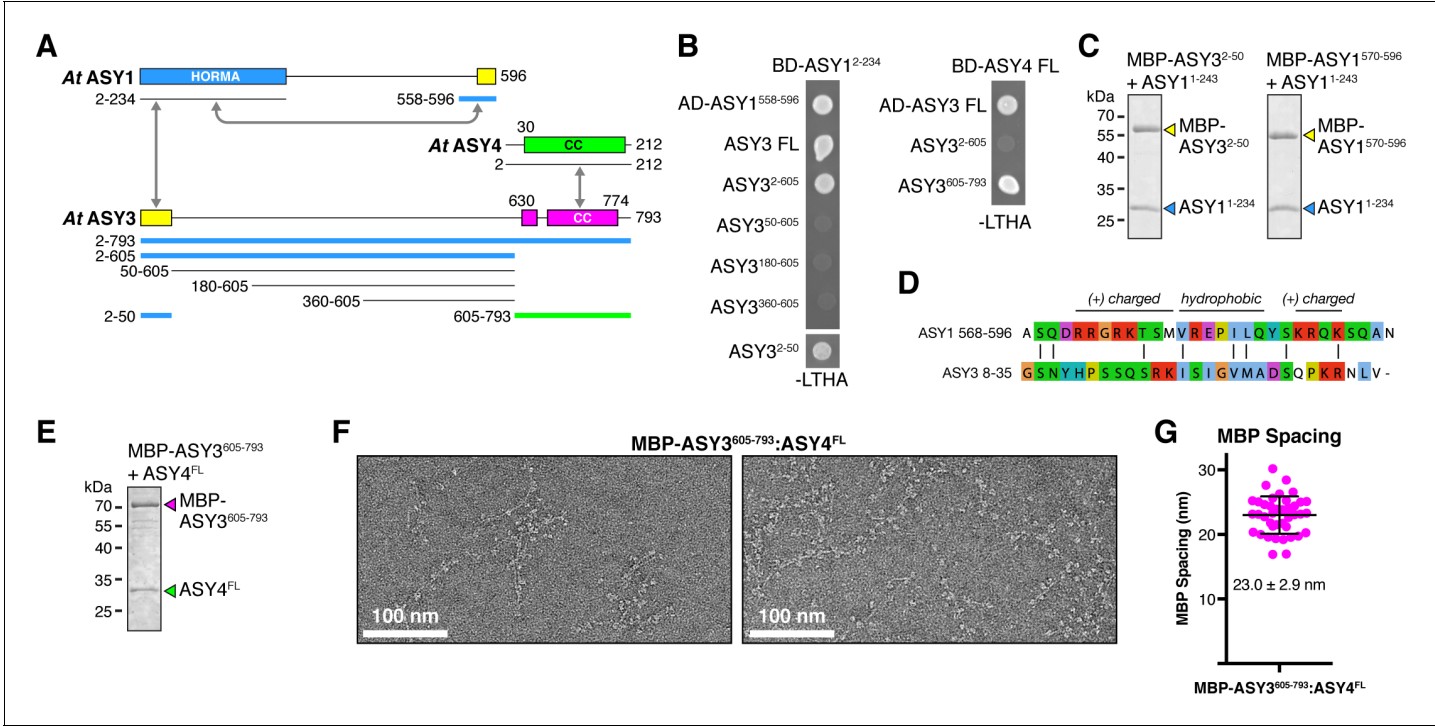

**Figure 5.** Plant ASY3 and ASY4 share a conserved molecular architecture with mammalian and budding-yeast axis proteins. (**A**) Schematic of *Arabidopsis thaliana* chromosome axis proteins, with truncations used for yeast two-hybrid assays shown for ASY1, ASY3, and ASY4. Colored in blue and green are ASY3 constructs that interact with the ASY1 HORMA domain (blue) and ASY4 (green). (**B**) Yeast two-hybrid analysis of *A. thaliana* ASY1, ASY3, and ASY4. AD: Gal4 activation domain fusion; BD: Gal4 DNA-binding domain fusion. Stringent selection on -LEU-TRP-HIS-ADE (-LTHA) media is shown; see *Figure 5—figure supplement 1* for complete results. (**C**) SDS-PAGE analysis of purified His$_6$-MBP-tagged closure motifs in ASY3 (residues 2–50) and ASY1 (residues 570–596) in complex with untagged ASY1 HORMA domain (residues 1–234). Complexes were purified using Ni$^{2+}$ affinity and size-exclusion chromatography. (**D**) Sequence alignment of the putative closure motif regions of *At* ASY1 (residues 568–596) and ASY3 (residues 8–35). The two regions show weak homology with a central region enriched in hydrophobic residues, bracketed on both sides by positively-charged residues. See *Figure 5—figure supplement 2* for sequence alignments of both regions. (**E**) SDS-PAGE analysis of purified His$_6$-MBP-ASY3$^{605-793}$:ASY4$^{FL}$ complexes used for negative-stain EM analysis (panel F). (**F**) Representative negative-stain electron micrographs of purified His$_6$-MBP-ASY3$^{605-793}$: ASY4$^{FL}$ filaments. See *Figure 5—figure supplement 3* for additional full micrographs. (**G**) Quantification of inter-MBP spacing in micrographs of His$_6$-MBP-*Mm* SYCP2$^{1325-1500}$:SYCP3$^{84-248}$ filaments. The measured spacing of 23.0 ± 2.9 nm (mean ±standard deviation from 41 measured intervals) is equivalent to the length of a ~160 residue coiled-coil (0.146 nm rise per residue). Predicted coiled-coil regions of ASY3 and ASY4 are ~145 and ~180 residues, respectively.

DOI: https://doi.org/10.7554/eLife.40372.019

The following figure supplements are available for figure 5:

**Figure supplement 1.** Yeast two-hybrid analysis of plant ASY1, ASY3, and ASY4.
DOI: https://doi.org/10.7554/eLife.40372.020

**Figure supplement 2.** Sequence alignments of putative closure motifs in plant ASY1 and ASY3.
DOI: https://doi.org/10.7554/eLife.40372.021

**Figure supplement 3.** Electron micrographs of *At* ASY3:ASY4 filaments.
DOI: https://doi.org/10.7554/eLife.40372.022

SYCP3$^{84-248}$ filament, these filaments were decorated at regular intervals with pairs of MBP densities. When we measured the average distance between paired MBP densities along these filaments, we obtained an average spacing of 23.0 ± 2.9 nm, in close agreement with the spacing in SYCP2:SYCP3 filaments and with the predicted length of the ASY3 and ASY4 coiled-coil regions (~145 and~180 residues, corresponding to coiled-coil lengths of ~21.2 and~26.3 nm; *Figure 5G*). These data strongly suggest that ASY3 and ASY4 assemble into 2:2 heterotetramers that associate end-to-end to form extended filaments, in a manner equivalent to both mammalian SYCP2:SYCP3 and fungal Red1.

## Discussion

The meiotic chromosome axis plays several crucial roles to support inter-homolog crossover formation and signaling in meiosis I. The first major role is to provide a scaffold for organization of each chromosome as a linear array of loops, with these loops directly extruded or otherwise constrained by cohesin complexes (*Zickler and Kleckner, 1999*). The axis assembles in early meiotic prophase, when chromosomes just become visible as the 'thin threads' for which the leptotene stage is named. As cells progress through zygotene and then pachytene ('thick threads'), chromosomes undergo significant linear compaction without disruption of the underlying chromosome axis structure. We have shown here that budding-yeast Red1, mammalian SYCP2:SYCP3, and plant ASY3:ASY4 all form filaments from homo- or hetero-tetrameric units, and that the SYCP2:SYCP3 filaments have a tendency to form bundles. We propose that individual short filaments of these 'axis core proteins' associate loosely with cohesin complexes, then form bundles to assemble a flexible scaffold for cohesin-mediated extrusion/constraint of chromatin loops (*Figure 6*). In this scheme, both filament formation by axis core proteins and cohesin activity are required for axis assembly and chromosome compaction, explaining how mutation of axis core proteins like SYCP3 (*Novak et al., 2008*; *Yuan et al., 2000*; *Yuan et al., 2002*) or cohesin subunits including SMC1β (*Novak et al., 2008*; *Revenkova et al., 2004*), REC8 (*Xu et al., 2005*), RAD21L (*Ward et al., 2016*), and STAG3 (*Fukuda et al., 2014*; *Hopkins et al., 2014*; *Ward et al., 2016*; *Winters et al., 2014*) can affect the overall length of the axis. An axis constructed from a flexible core of loosely-associated filaments would also enable axis extension or compression in processes like synaptic adjustment, in which the lengths of two chromosomes can adjust to one another as the synaptonemal complex (SC) assembles between them (*Zickler and Kleckner, 1999*).

A second critical function of the chromosome axis is to promote the formation of meiotic DSBs, then orchestrate the repair of a subset of DSBs as inter-homolog crossovers. In most organisms, the meiotic HORMADs are major regulators of both DSB formation and crossover formation. We have previously proposed an overall axis assembly pathway in *S. cerevisiae* with cohesin-associated Red1 recruiting Hop1 through its closure motif (*West et al., 2018*), and we can now extend this model to both mammals and plants. We propose that a key conserved function of the axis core proteins is to recruit meiotic HORMADs through their HORMA domain-binding closure motifs. These localized HORMADs may then recruit additional HORMADs through head-to-tail oligomer formation through their own C-terminal closure motifs. Finally, as cells enter pachytene and the axis core proteins become integrated into the SC, HORMADs are removed from the axis by the Pch2/TRIP13 ATPase, thereby suppressing further DSB formation and licensing the progression of meiosis (*Börner et al., 2008*; *Joshi et al.,*

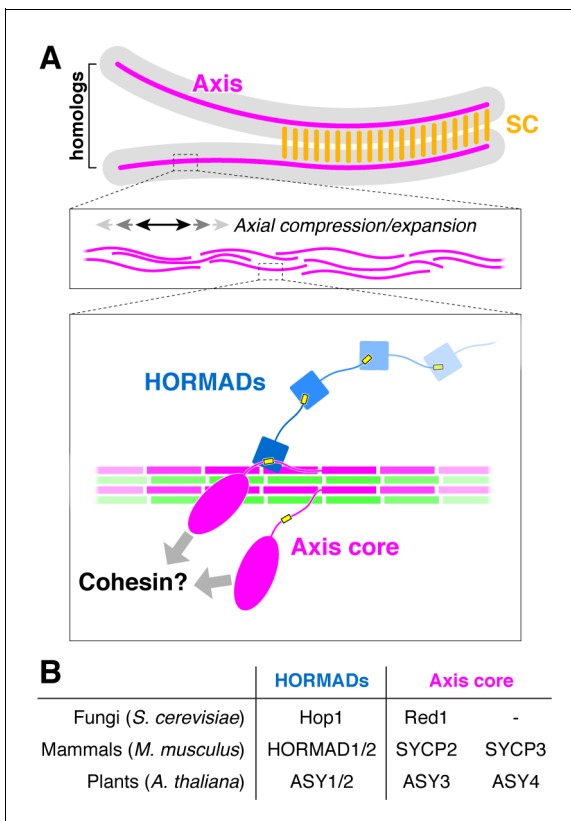

**Figure 6.** The conserved molecular architecture of the meiotic chromosome axis. (**A**) Model for assembly of the meiotic chromosome axis in fungi, plants, and mammals. Related axis core proteins (fungal Red1, plant ASY3:ASY4, mammalian SYCP2:SYCP3) form filaments from coiled-coil homo- or heterotetrameric units, which flexibly associate with chromosome-associated cohesin complexes. Chromatin loop extrusion by cohesin complexes and bundling of axis-core filaments leads to formation of the chromosome axis, which is flexible and able to axially compress or expand if needed. (**B**) List of homologous chromosome axis proteins in different eukaryotic groups.
DOI: https://doi.org/10.7554/eLife.40372.023

*2009*; *Roig et al., 2010*; *Smith and Roeder, 1997*; *Wojtasz et al., 2009*).

Importantly, while we show that HORMAD recruitment by axis core proteins is conserved across fungi, mammals, and plants, additional architectural complexity likely exists in organisms with multiple HORMAD proteins. For example, mammalian HORMAD1 and HORMAD2 play distinct roles in meiotic regulation, and may also have distinct recruitment mechanisms: we have demonstrated an interaction between SYCP2 and HORMAD2, but not HORMAD1, leaving open the possibility that HORMAD1 is recruited by other means. In plants, the two HORMAD proteins ASY1 and ASY2 may similarly possess distinct recruitment mechanisms to drive differential biological functions. Further work outlining the specific recruitment mechanisms of individual HORMADs, including their potential dependence on one another through head-to-tail oligomer assembly, will be required to fully understand chromosome axis architecture and function in these organisms.

The third major function of the meiotic axis is to serve as the lateral elements of the SC in pachytene, after the bulk of meiotic HORMADs have been removed. Our physical model of the mammalian chromosome axis, comprising a bundle of SYCP2:SYCP3 filaments with a periodicity of 23 nm, generally agrees with prior electron microscopy (EM) studies showing ~20 nm periodicity in the lateral elements of assembled SCs (*Ortiz et al., 2002*). Also in agreement with this model, a recent analysis of mouse SC structure by super-resolution light microscopy has shown that SYCP3 and the SYCP2 coiled-coil region perfectly co-localize in a single 'cable' in each lateral element, and that the C-terminus of the transverse filament protein SYCP1 is situated close to this cable (*Schücker et al., 2015*). Significant questions remain regarding how the SC lateral elements and transverse filaments might interact, and the role of cohesin complexes in this interaction is also unknown. Recently, it was reported that REC8 and RAD21L, meiosis-specific cohesin complex kleisin subunits, both localize slightly 'inside' SYCP3; that is, they are situated closer to the SYCP1 transverse filaments than the SYCP2:SYCP3 complex (*Rong et al., 2016*). These data suggest that cohesin complexes may somehow be integrated into the structure of the SC, in a manner that is not yet well-understood.

Several recent studies have reported that in vitro, mammalian SYCP3 forms coiled-coil homotetramers that further assemble into large oligomeric structures with a 22 nm periodicity (*Syrjänen et al., 2014*). Further, SYCP3 can bind DNA through two short patches of basic residues near the N-terminus of this protein's coiled-coil domain (*Syrjänen et al., 2014*), and large SYCP3 oligomers appear to bind and condense plasmid DNA (*Bollschweiler et al., 2018*). These data have led to a model whereby homotypic SYCP3 oligomers, interacting directly with DNA, form a major part of the mammalian chromosome axis (*Syrjänen et al., 2014*). While we cannot rule out the formation of homotypic SYCP3 assemblies in meiotic cells, our data shows that the SYCP2:SYCP3 heterotetramer is more stable in solution than the SYCP3 homotetramer, and is therefore likely to be the preferred assembly in meiotic cells. Further, as SYCP3 does not localize to the chromosome axis in a mutant of SYCP2 lacking its coiled-coil domain (*Yang et al., 2006*), direct SYCP3-DNA binding is unlikely to contribute significantly to axis formation.

We have shown that the architecture of the meiotic chromosome axis is highly conserved across fungi, mammals, and plants. Our model assigns critical functions in both overall axis structure and HORMAD recruitment to the axis core proteins, yet some organisms, including *C. elegans* and *D. melanogaster*, appear to lack axis core proteins entirely. Our prior work has strongly suggested that the meiotic HORMADs in *C. elegans* interact directly with cohesin complexes (*Kim et al., 2014*), and thus far meiotic HORMADs have not been identified in *D. melanogaster*. We propose that the key feature of meiosis in both *C. elegans* and *D. melanogaster* that eliminates the need for axis core proteins is that these organisms assemble the SC prior to meiotic recombination. Thus, the SC itself can provide a physical scaffold for chromosome organization and recombination control in these organisms, eliminating much of the need for a distinct chromosome axis that assembles prior to SC formation.

While our data shed significant new light on the assembly and function of the meiotic chromosome axis, significant questions remain. First, while the N-terminal domains of both Red1 and SYCP2 likely adopt similar structures and mediate these proteins' association with meiotic chromosomes, their direct binding partners are as yet mysterious. A recent study identified several potential binding partners of the SYCP2 N-terminal domain (*Feng et al., 2017*), but as these proteins are mostly centromere-associated, it remains unknown what SYCP2 may bind along the length of chromosomes. We propose that the SYCP2 and Red1 N-terminal domains may bind cohesin complexes

directly (*Sun et al., 2015*), or may instead bind one or more chromatin-associated proteins, perhaps even a specific histone mark, to mediate a flexible interaction with chromatin.

A further mystery involves plant ASY3, which appears to entirely lack a Red1/SYCP2-like N-terminal domain. This protein may associate with chromosome-localized proteins through one or more short conserved motifs in its extended disordered region, or may in fact be recruited through interactions with the HORMADs ASY1 and ASY2. Both plant ASY1 and fungal Hop1 possess putative DNA- or protein-binding domains in their central regions, raising the possibility that these meiotic HORMADs could recruit axis core proteins to meiotic chromosomes, in a reversal of the canonical localization-dependence of these proteins. Thus, while the overall theme of axis assembly through filament formation is likely conserved across many eukaryotic families, each family appears to have evolved variations on this theme in keeping with its own unique requirements.

# Materials and methods

**Key resources table**

| Reagent type or resource | Designation | Source or reference | Identifiers | Additional information |
|---|---|---|---|---|
| Strain, strain background (*S. cerevisiae*) | strain AH109 | Clontech | | |
| Strain, strain background (*S. cerevisiae*) | strain Y187 | Clontech | | |
| Strain, strain background (*E. coli*) | Rosetta2 (DE3) pLysS | Novagen | | |
| Genetic reagent (*S. cerevisiae*) | S288c genomic DNA | Invitrogen | cat. # 69240 | |
| Genetic reagent (*Z. rouxii*) | NRRL Y-229 genomic DNA | ATCC | cat. # 2623D-5 | |
| Genetic reagent (*M. musculus*) | HORMAD1 cDNA | TransOMIC Technologies | BC051129 | |
| Genetic reagent (*M. musculus*) | HORMAD2 cDNA | TransOMIC Technologies | BC120781 | |
| Genetic reagent (*M. musculus*) | SYCP2 cDNA | Harvard PlasmID | MmCD0083242 | |
| Genetic reagent (*M. musculus*) | SYCP3 | GeneArt | | Synthesized gene fragment |
| Genetic reagent (*H. sapiens*) | SYCP2 | GeneArt | | Synthesized gene fragment |
| Genetic reagent (*H. sapiens*) | SYCP3 | GeneArt | | Synthesized gene fragment |
| Genetic reagent (*A. thaliana*) | ASY1 | GeneArt | | Synthesized gene fragment |
| Genetic reagent (*A. thaliana*) | ASY3 | GeneArt | | Synthesized gene fragment |

*Continued on next page*

*Continued*

| Reagent type or resource | Designation | Source or reference | Identifiers | Additional information |
|---|---|---|---|---|
| Genetic reagent (*A. thaliana*) | ASY4 | GeneArt | | Synthesized gene fragment |
| Antibody | Rabbit polyclonal Anti-Red1 antibody (1) | Gift from G. S. Roeder | | used for immunofluorescence |
| Antibody | Rabbit polyclonal Anti-Red1 antibody (2) | Gift from N. Hollingsworth | | used for Western blotting |
| Antibody | Mouse polyclonal Anti-Gmc2 antibody | Prosci Inc. | | used for immunofluorescence |
| Antibody | Goat polyclonal anti-rabbit HRP antibody | Jackson Immunoresearch | 111-035-003 | used for Western blotting |
| Recombinant DNA reagent | Plasmid pGADT7 | Clontech | | |
| Recombinant DNA reagent | Plasmid pBridge | Clontech | | |
| Recombinant DNA reagent | Macrolab vector 2CT | UC Berkeley Macrolab | | |
| Recombinant DNA reagent | Macrolab vector 13S-A | UC Berkeley Macrolab | | |
| Recombinant DNA reagent | Macrolab vector 2 ST | UC Berkeley Macrolab | | |
| Peptide, recombinant protein | TEV protease | David Waugh, National Cancer Institute | Clone pRK793 | Purified in-house |
| Peptide, recombinant protein | Mm HORMAD2 closure motif peptide | Biomatik, Inc. | | Sequence: FITC-Ahx-EPSFES SRKKKVS EPVTVFIPNRK |
| Commercial assay or kit | HisTrap HP column | GE Life Sciences | | |
| Commercial assay or kit | HiTrap Q HP column | GE Life Sciences | | |
| Commercial assay or kit | Superdex 200 column | GE Life Sciences | | |
| Chemical compound, drug | D0/D12 BS3 (bis-sulfosuccini midylsuberate | Creative Biomolecules | cat. # BS3 | |
| Software, algorithm | RAPD | https://github.com/RAPD | | |
| Software, algorithm | AIMLESS | http://www.ccp4.ac.uk | | |
| Software, algorithm | TRUNCATE | http://www.ccp4.ac.uk | | |
| Software, algorithm | autoxds | Stanford Synchrotron Radiation Lightsource | | In-house script |
| Software, algorithm | ARCIMBOLDO | http://chango.ibmb.csic.es | | |
| Software, algorithm | PHENIX | http://www.phenix-online.org/download/ | | |
| Software, algorithm | PHASER | http://www.phaser.cimr.cam.ac.uk/index.php/Phaser_Crystallographic_Software | | |

*Continued*

| Reagent type or resource | Designation | Source or reference | Identifiers | Additional information |
|---|---|---|---|---|
| Software, algorithm | RESOLVE | https://solve.lanl.gov | | |
| Software, algorithm | COOT | https://www2.mrc-lmb.cam.ac.uk/personal/pemsley/coot/ | | |
| Software, algorithm | ATSAS | https://www.embl-hamburg.de/biosaxs/software.html | | SAXS analysis suite |
| Software, algorithm | xQuest | http://prottools.ethz.ch/orinner/public/htdocs/xquest/ | | |
| Software, algorithm | Graphpad Prism | Graphpad software - https://www.graphpad.com | version 7 | |

## Cloning and protein purification

### Mammalian proteins

For yeast two-hybrid analysis, *M. musculus* genes were PCR-amplified from cDNA (SYCP2: Harvard PlasmID clone MmCD00083242; HORMAD1: TransOMIC technologies clone BC051129; HORMAD2: TransOMIC technologies clone BC120781) or synthesized DNA fragment (SYCP3; GeneArt) and inserted by ligation-independent cloning into modified pBridge and pGADT7 vectors (Clontech). For co-expression, *M. musculus* SYCP2 and SYCP3 fragments were separately cloned into UC Berkeley Macrolab vectors 2CT (SYCP2; $Amp^R$, N-terminal $His_6$-MBP fusion) or 13S-A (SYCP3; $Spec^R$, no tag) by ligation-independent cloning. For expression of SYCP3 alone, *M. musculus* SYCP3[105-248] was cloned into UC Berkeley Macrolab vector 2 ST (N-terminal $His_6$-SUMO fusion). The *H. sapiens* SYCP2-SYCP3 fusion construct was assembled by multi-part PCR from synthesized-fragment templates (GeneArt) and inserted into vector 2CT by ligation-independent cloning. For co-expression of *M. musculus* SYCP2:HORMAD2 complexes, a polycistronic expression cassette was assembled by PCR and inserted into vector 2CT, yielding a final vector encoding $His_6$-MBP-tagged SYCP2 fragments plus untagged HORMAD2[1-241].

For purification of SYCP2:SYCP3 complexes, $His_6$-MBP-SYCP2 and SYCP3 constructs were co-transformed into *E. coli* strain Rosetta 2(DE3) pLysS (Novagen), and grown in the presence of ampicillin, spectinomycin and chloramphenical to an $OD_{600}$ of 0.9 at 37°C, induced with 0.25 mM IPTG, then grown for a further 16 hr at 18°C prior to harvesting by centrifugation. For purification, cells were lysed by sonication, then clarified lysates were purified by $Ni^{2+}$ affinity (HisTrap HP; GE Life Sciences), ion exchange (HiTrap SP or Q; GE Life Sciences), and size exclusion chromatography (Superdex 200; GE Life Sciences). *H. sapiens* SYCP2:HORMAD2 complexes were coexpressed as above, then purified by $Ni^{2+}$ affinity chromatography and analyzed by SDS-PAGE. In cases where cleavage of N-terminal $His_6$-MBP or $His_6$-SUMO tags was required, tags were removed by incubation with TEV protease at 4°C for 16 hr, then the mixture was passed over a $Ni^{2+}$ affinity column, and the flow-through fractions were concentrated and purified by size-exclusion chromatography.

For size-exclusion chromatography-based assays of SYCP2:SYCP3 filament formation, $His_6$-MBP-SYCP2:SYCP3 complexes were initially purified by Ni-NTA chromatography, then passed over a Superose-6 size-exclusion column (GE Life Sciences) to remove small-molecular weight contaminants. Fractions corresponding to the entire range containing SYCP2:SYCP3 complexes were pooled, concentrated, then passed over Superose-6 a second time for the traces shown in *Figure 3B*. N-terminal $His_6$-MBP tags were not removed for this analysis.

For size exclusion chromatography coupled to multi-angle light scattering (SEC-MALS), 100 µL purified proteins at 2–5 mg/mL was injected onto a Superdex 200 Increase 10/300 GL column (GE Life Sciences) in a buffer containing 20 mM HEPES pH 7.5, 300 mM NaCl, 5% glycerol, and 1 mM DTT. Light scattering and refractive index profiles were collected by miniDAWN TREOS and Optilab T-rEX detectors (Wyatt Technology), respectively, and molecular weight was calculated using ASTRA v. six software (Wyatt Technology).

### Fungal proteins

*Zygosaccharomyces rouxii* Red1 constructs were amplified by PCR and inserted by ligation-independent cloning into UC Berkeley Macrolab vector 2CT (AmpR, N-terminal His$_6$-MBP fusion) for expression in *E. coli*. Proteins were expressed and purified as above.

### Plant proteins

Full-length codon-optimized genes for *Arabidopsis thaliana* ASY1, ASY3, and ASY4 were synthesized (GeneArt) and inserted by ligation-independent cloning into modified pBridge and pGADT7 vectors (Clontech) for yeast two-hybrid analysis, or cloned into UC Berkeley Macrolab vectors 2CT/13S-A for expression in *E. coli*. Truncations were amplified by PCR and similarly cloned.

For co-purification of the ASY1 HORMA domain with putative closure motif peptides, putative closure motifs (ASY3$^{2-50}$ and ASY1$^{570-596}$) in vector 2CT (N-terminal His$_6$-MBP fusion) and ASY1$^{1-234}$ in vector 13S-A (untagged) were co-transformed into *E. coli* strain Rosetta 2(DE3) pLysS, and grown in the presence of ampicillin, spectinomycin and chloramphenical to an OD$_{600}$ of 0.9 at 37°C, induced with 0.25 mM IPTG, then grown for a further 16 hr at 18°C prior to harvesting by centrifugation. For purification, cells were lysed by sonication, then clarified lysates were purified by Ni$^{2+}$ affinity (HisTrap HP; GE Life Sciences) and size exclusion chromatography (Superdex 200; GE Life Sciences). For purification of ASY3:ASY4 for electron microscopy, ASY3$^{605-793}$ in vector 2CT (N-terminal His$_6$-MBP fusion) and full-length ASY4 in vector 13S-A (untagged) were co-transformed into *E. coli* strain Rosetta 2(DE3) pLysS, and grown in the presence of ampicillin, spectinomycin and chloramphenical to an OD$_{600}$ of 0.9 at 37°C, induced with 0.25 mM IPTG, then grown for a further 16 hr at 18°C prior to harvesting by centrifugation, and purified as above.

## Yeast two-hybrid

For yeast two-hybrid analysis, plasmids were transformed into AH109 and Y187 yeast strains (Clontech), and transformants were selected using CSM -Leu (for pGADT7 vectors) and CSM -Trp (pBridge vectors) media. Haploid yeast strains were mated overnight at room temperature, and diploids were selected using CSM -Leu-Trp media. Diploids were patched onto low-stringency (CSM -Leu-Trp-His) and high stringency media (CSM -Trp-Leu-His-Ade), grown for 1–3 days at 30°C, and imaged.

## Fluorescence polarization

An N-terminal FITC-Ahx labeled *Mm* HORMAD2$^{288-306}$ peptide was synthesized (BioMatik), resuspended in DMSO, then diluted into binding buffer (20 mM Tris pH 7.5, 300 mM NaCl, 10% glycerol, 1 mM DTT, 0.1% NP-40). Fifty µL reactions containing 50 nM peptide plus up to 50 µM bait proteins were incubated 60 min at room temperature, then fluorescence polarization was read in 384-well plates using a TECAN Infinite M1000 PRO fluorescence plate reader. All binding curves were done in triplicate. Binding data were analyzed with Graphpad Prism v. seven using a single-site binding model.

## Electron microscopy

For negative-stain electron microscopy, protein complexes were passed over a size exclusion column (Superdex 200 Increase 10/300 GL; GE Life Sciences) in EM buffer (300 mM NaCl, 20 mM Tris-HCl pH 7.5, 1 mM DTT), and peak fractions were diluted to ~0.01 mg/mL in EM buffer. Samples were spotted on freshly glow-discharged carbon coated copper grids, blotted into a thin film, and stained using 2% of uranyl formate. Electron micrographs were acquired on a Tecnai F20 Twin transmission electron microscope (FEI, Hillsboro OR) operating at 200 kV on a Tietz F416 4K × 4K CMOS camera (TVIPS, Gauting, Germany). For untagged *Zr* Red1$^{705-798}$ and MBP-ASY3$^{605-793}$:ASY4$^{FL}$, micrographs were acquired on a FEI Talos F200C with 4K × 4K CMOS camera (Thermo Fisher Scientific). Micrographs of His$_6$-MBP-SYCP2$^{1325-1500}$:SYCP3$^{84-248}$ and MBP-ASY3$^{605-793}$:ASY4$^{FL}$ were analyzed using ImageJ to determine the average spacing of MBP densities on the respective filaments.

## Thermofluor melting assays

For measurement of melting temperature, 45 uL 0.1 mg/mL purified protein in gel-filtration buffer (20 mM Tris-HCl pH 7.5, 300 mM NaCl, 10% glycerol, 1 mM DTT) was mixed with 5 uL 50X SYPRO

orange dye (Life Technologies; 5X final concentration) and pipetted into an optically-clear qPCR plate. SYPRO fluorescence was measured in a Bio-Rad CFX96 qPCR machine in FRET mode (excitation 450–490 nm, emission 560–580) using a temperature range 25–95°C in 0.5° steps (15 s hold per step). Triplicate measurements were averaged, buffer-subtracted, then the derivative of the fluorescence was calculated. The maximum value of the derivative curve (highest rate of change in fluorescence) is assigned as the $T_m$. N-terminal His$_6$-MBP and His$_6$-SUMO on SYCP2$^{1325-1479}$:SYCP3$^{105-248}$ and SYCP3$^{105-248}$, respectively, were removed prior to Tm analysis.

## Crystallization and structure determination of *M. musculus* SYCP3 homotetramer

When co-expressed in *E. coli*, *M. musculus* SYCP3 is expressed at much higher levels than SYCP2 (not shown). We found that while *M. musculus* SYCP2$^{CC}$ is insoluble when expressed without SYCP3$^{CC}$, SYCP3$^{CC}$ is able to form soluble homotetramers. While optimizing expression constructs, we co-expressed *M. musculus* His$_6$-SUMO-SYCP3$^{105-248}$ with untagged SYCP2$^{1325-1472}$, purified the resulting complex, and identified crystallization conditions. Crystals were obtained in hanging drop format by mixing protein (50–80 mg/mL) with two parts well solution containing 100 mM Tris-HCl pH 8.5, 16% PEG 4000, and 100–200 mM sodium acetate. Later analysis showed that these crystals contain SYCP3 homotetrameric complexes, rather than SYCP2:SYCP3 heterotetramers. Because of the tendency of SYCP3$^{CC}$ to form homotetrameric complexes, all other analysis with SYCP2$^{CC}$:SYCP3$^{CC}$ complexes was performed with complexes expressed with tagged SYCP2 and untagged SYCP3.

SYCP3 homotetramer crystals were cryoprotected by the addition of 20% sucrose, then diffraction data was collected at the Advanced Photon Source, beamline 24ID-C. Despite identical growth conditions and similar shape, crystals belonged to two different space groups (P1 and P2$_1$; *Supplementary file 1*). Data collected at the Advanced Photon Source were indexed and scaled by RAPD (https://github.com/RAPD), which used XDS (*Kabsch, 2010*) for indexing and data reduction, and the CCP4 programs AIMLESS (*Evans and Murshudov, 2013*) and TRUNCATE (*Winn et al., 2011*) for scaling and conversion to structure factors. Data collected at the Stanford Synchrotron Radiation Lightsource was indexed and scaled by the *autoxds* script, which uses XDS, AIMLESS, and TRUNCATE as above. An initial model was determined by ARCIMBOLDO_LITE (*Sammito et al., 2015*) in its COILED_COIL mode (*Caballero et al., 2018*) using a merged P2$_1$ dataset assembled from three individual datasets from different crystals, cut to a final resolution of 2.5 Å. ARCIMBOLDO (*Millán et al., 2015*) uses PHASER (*McCoy et al., 2007*) to place individual α-helices by eLLG (expected log likelihood-gain)-guided molecular replacement (*Oeffner et al., 2018*), then expand partial solutions with SHELXE (*Usón and Sheldrick, 2018*) through density modification and autotracing into a complete model (*Usón et al., 2007*). Phases from the initial ARCIMBOLDO model (393 residues) were used to identify selenomethionine sites, which were then supplied to the Phenix Autosol module (*Terwilliger et al., 2009*) for phase calculation in PHASER (*McCoy et al., 2007*; *Read and McCoy, 2011*), density modification including two-fold NCS averaging in RESOLVE (*Terwilliger, 2003*), and initial model building in RESOLVE. Initial models from ARCIMBOLDO and RESOLVE were manually rebuilt in COOT and refined in phenix.refine (*Adams et al., 2010*) against a single 2.5 Å-resolution dataset collected from crystals of selenomethionine-substituted protein. The register of all four protein chains in the final model, and their identity as SYCP3$^{CC}$, were verified by anomalous difference maps showing the location of selenomethionine residues. While ARCIMBOLDO successfully determined the structure in the P1 crystal form, the initial P1 model used for rebuilding and refinement was generated by molecular replacement in PHASER using the P2$_1$ model. The P1 model was refined against a 2.2 Å-resolution dataset generated by merging five independent datasets collected at APS beamline 24ID-E and SSRL beamline 14–1.

## Support statement - Advanced Photon Source NE-CAT beamline 24ID-C

This work is based upon research conducted at the Northeastern Collaborative Access Team beamlines, which are funded by the National Institute of General Medical Sciences from the National Institutes of Health (P41 GM103403). The Pilatus 6M detector on 24-ID-C beam line is funded by a NIH-ORIP HEI grant (S10 RR029205). This research used resources of the Advanced Photon Source, a U.

S. Department of Energy (DOE) Office of Science User Facility operated for the DOE Office of Science by Argonne National Laboratory under Contract No. DE-AC02-06CH11357.

## Support statement - Stanford Synchrotron Radiation Lightsource beamline 14–1

Use of the Stanford Synchrotron Radiation Lightsource, SLAC National Accelerator Laboratory, is supported by the U.S. Department of Energy, Office of Science, Office of Basic Energy Sciences under Contract No. DE-AC02-76SF00515. The SSRL Structural Molecular Biology Program is supported by the DOE Office of Biological and Environmental Research, and by the National Institutes of Health, National Institute of General Medical Sciences (including P41GM103393). The contents of this publication are solely the responsibility of the authors and do not necessarily represent the official views of NIGMS or NIH.

## Small-angle X-ray scattering (SAXS)

For SAXS, $Mm$ SYCP2$^{CC}$:SYCP3$^{CC}$ was diluted to 1, 3, or 6 mg/mL in a buffer containing 20 mM Tris-HCl pH 8.5, 300 mM NaCl, 2% glycerol, and 1 mM DTT. Both His$_6$-MBP-tagged and untagged $Hs$ SYCP3$^{CC}$-SYCP2$^{CC}$ fusion was diluted to 2, 4 or 8 mg/mL in a buffer containing 20 mM Tris-HCl pH 7.5, 300 mM NaCl, 2% glycerol, and 1 mM DTT. SAXS data were collected at the SIBYLS Beamline 12.3.1 at the Advanced Light Source, Lawrence Berkeley National Lab (which is funded by DOE BER Integrated Diffraction Analysis Technologies (IDAT) program and NIGMS grant P30 GM124169-01, ALS-ENABLE) (*Dyer et al., 2014*). For each sample, thirty 0.3 s exposures were taken and integrated, for a total exposure time of 10 s. Exposures were radially averaged and buffer-subtracted to yield SAXS scattering curves. SAXS data analysis was performed with ScÅtter (https://bl1231.als.lbl.gov/scatter/) and the ATSAS SAXS analysis suite (https://www.embl-hamburg.de/biosaxs/software.html) (*Dyer et al., 2014*).

## Crosslinking mass spectrometry (XLMS)

For cross-linking of $Hs$ SYCP3$^{CC}$-SYCP2$^{CC}$, the protein was diluted to 1 mg/mL in a buffer containing 20 mM HEPES pH 7.5, 300 mM NaCl, 10% glycerol, and 1 mM DTT. Crosslinking was performed by addition of 0.2, 0.5, or 1 mM isotopically-coded D$_0$/D$_{12}$ BS$^3$ (bis-sulfosuccinimidylsuberate; Creative Molecules) for 60 min at room temperature. The reaction was quenched by the addition of 100 mM NH$_4$HCO$_3$ and further incubation at 30°C for 10 min. Quenched reactions were supplemented with 8M urea to a final concentration of 6M. Subsequent to reduction and alkylation, crosslinked proteins were digested with Lys-C (1:50 w/w, Wako) for 3 hr, diluted with 50 mM ammonium bicarbonate to 1M urea and digested with trypsin (1:50 w/w, Promega) overnight. Crosslinked peptides were purified by reversed phase chromatography using C18 cartridges (Sep-Pak, Waters). Crosslink fractions by peptide size exclusion chromatography and analyzed by tandem mass spectrometry (Orbitrap Elite, Thermo Scientific) (*Herzog et al., 2012*). Fragment ion spectra were searched and crosslinks identified by the dedicated software program xQuest (*Walzthoeni et al., 2015*). All unique detected crosslinks are listed in *Supplementary file 2* and *Supplementary file 3*.

## Yeast genetics and imaging

All yeast strains were derived from the SK1-related diploid strain NH144 (*Supplementary file 5*) (*de los Santos and Hollingsworth, 1999*; *Hollingsworth et al., 1995*). For *Sc-Zr* Red1 chimeras, a homologous recombination template was generated to replace residues 734–827 with residues 705–798 (wild-type or I715R) or 705–791 of *Zr* Red1, followed by a *KanMX* selection marker, and integrated into the *RED1* locus. For spore viability, cells were grown on YPD agar, patched onto SPO medium (1% KOAc) for 48–72 hr, then tetrads were dissected onto YPD agar and grown 3 days for analysis. Spore viability was 95.3% (122 viable spores out of 128) for *RED1*, 54% for *red1-Sc$^{1-734}$:Zr$^{707-798}$* (28 viable out of 52), 2% for *red1-Sc$^{1-734}$:Zr$^{707-791}$* (1 viable out of 56), and 9.4% for *red1-Sc$^{1-734}$:Zr$^{707-798}$I715R]* (12 viable out of 128).

For synchronous meiosis and fluorescence imaging, cells were sporulated as in (*Subramanian et al., 2016*). Briefly, cells were grown in YPD, then diluted into BYTA (BYTA; 50 mM sodium phthalate-buffered, 1% yeast extract, 2% tryptone and 1% acetate) at OD$_{600}$ = 0.3, grown overnight, then washed and resuspended in SPO medium (0.3% KOAc pH 7.0) at OD$_{600}$ = 2.0 at 30°

C to induce sporulation. Samples were removed at 3 and 5 hr after transfer to SPO medium, then meiotic nuclei were surface-spread on glass slides and fixed as previously described (*Voelkel-Meiman et al., 2016*), then imaging was carried out using a Deltavision RT Imaging System (Applied Precision) adapted to an Olympus (IX71) microscope. Cells were stained with DAPI, anti-Red1, and anti-Gmc2. Polyclonal mouse anti-Gmc2 antibodies were raised against purified Gmc2 protein (Pro-Sci Inc.); this antibody was used at 1:800 dilution. Polyclonal rabbit anti-Red1 (a kind gift from GS Roeder, *Smith and Roeder, 1997*) was used at 1:100 dilution. Secondary antibodies conjugated with Alexa Fluor dyes were purchased from Jackson ImmunoResearch and used at 1:200 dilution. For quantification of Red1 and Gmc2 spatial distribution on meiotic chromosomes, 30–50 nuclei were manually scored per condition.

For western blotting, we used primary rabbit anti-Red1 (a kind gift from Nancy Hollingsworth) at 1:10,000 dilution, and secondary goat anti-rabbit HRP (Jackson Immunoresearch) at 1–10,000 dilution.

## Acknowledgements

The authors thank the staffs of the Stanford Synchrotron Light Source and the Advanced Photon Source sector 24 for assistance with collecting x-ray diffraction data, the staff of the Advanced Light Source Sibyls Beamline 12.3.1 for assistance with SAXS data collection and processing, and T Booth at the UCSD Cryo-Electron Microscopy Facility for assistance with electron microscopy. We thank members of the Corbett lab, A Desai, and Y Kim for critical reading and helpful discussions. SU acknowledges past support from the UC San Diego Molecular Biophysics Training Grant (National Institutes of Health T32 GM008326), and current support from the National Science Foundation (Graduate Research Fellowship). IU and IC are supported by grants BIO2015-64216-P and MDM2014-0435 (the Spanish Ministry of Science, Innovation and Universities). AJM acknowledges support from the National Institutes of Health (R15 GM116109). KDC acknowledges past support from the Ludwig Institute for Cancer Research and the National Institutes of Health (R01 GM104141). KDC and FH acknowledge joint support from the Human Frontiers Science Program (RGP0008/2015).

## Additional information

### Funding

| Funder | Grant reference number | Author |
|---|---|---|
| National Institutes of Health | R01 GM104141 | Alan MV West<br>Scott C Rosenberg<br>Madison K Lehmer<br>Qiaozhen Ye<br>Kevin D Corbett |
| National Science Foundation | Graduate Research Fellowship | Sarah N Ur |
| National Institutes of Health | T32 GM008326 | Sarah N Ur |
| Spanish Ministry of Science and Innovation | BIO2015-64216-P | Iracema Caballero<br>Isabel Usón |
| Spanish Ministry of Science and Innovation | MDM2014-0435 | Iracema Caballero<br>Isabel Usón |
| National Institutes of Health | R15 GM116109 | Amy J MacQueen |
| Human Frontier Science Program | RGP0008/2015 | Franz Herzog<br>Kevin D Corbett |
| Ludwig Institute for Cancer Research | | Kevin D Corbett |

The funders had no role in study design, data collection and interpretation, or the decision to submit the work for publication.

## Author contributions
Alan MV West, Scott C Rosenberg, Conceptualization, Investigation, Visualization, Writing—original draft, Writing—review and editing; Sarah N Ur, Madison K Lehmer, Qiaozhen Ye, Investigation, Visualization, Writing—review and editing; Götz Hagemann, Iracema Caballero, Software, Investigation, Writing—review and editing; Isabel Usón, Software, Supervision, Investigation, Writing—review and editing; Amy J MacQueen, Supervision, Investigation, Visualization, Writing—review and editing; Franz Herzog, Software, Formal analysis, Supervision, Funding acquisition, Investigation, Writing—review and editing; Kevin D Corbett, Conceptualization, Supervision, Funding acquisition, Investigation, Visualization, Methodology, Writing—original draft, Project administration, Writing—review and editing

## Author ORCIDs
Franz Herzog ⓘ http://orcid.org/0000-0001-8270-1449
Kevin D Corbett ⓘ http://orcid.org/0000-0001-5854-2388

## Decision letter and Author response
Decision letter https://doi.org/10.7554/eLife.40372.039
Author response https://doi.org/10.7554/eLife.40372.040

# Additional files

## Supplementary files
• Supplementary file 1. Data collection and refinement statistics.
DOI: https://doi.org/10.7554/eLife.40372.024

• Supplementary file 2. SYCP3-SYCP2 crosslinks.
DOI: https://doi.org/10.7554/eLife.40372.025

• Supplementary file 3. SYCP3-SYCP3 and SYCP2-SYCP2 crosslinks.
DOI: https://doi.org/10.7554/eLife.40372.026

• Supplementary File 4. Simulated versus experimental crosslinking in parallel vs. antiparallel models of tetramer assembly.
DOI: https://doi.org/10.7554/eLife.40372.027

• Supplementary file 5. Yeast strains.
DOI: https://doi.org/10.7554/eLife.40372.028

• Transparent reporting form
DOI: https://doi.org/10.7554/eLife.40372.029

## Data availability
Primary diffraction data for *M. musculus* SYCP3 tetramer structures have been deposited with the SBGrid Data Bank (https://data.sbgrid.org) under dataset numbers 583 (P21 crystal form) and 584 (P1 form). Reduced diffraction data and refined structural models have been deposited with the Protein Data Bank (www.pdb.org) under accession numbers 6DD8 (P21 crystal form) and 6DD9 (P1 form)

The following datasets were generated:

| Author(s) | Year | Dataset title | Dataset URL | Database and Identifier |
|---|---|---|---|---|
| Rosenberg SC, Munoz IC | 2018 | Structure of mouse SYCP3, P21 form | https://www.rcsb.org/structure/6DD8 | Protein Data Bank, 6DD8 |
| Rosenberg SC | 2018 | Structure of mouse SYCP3, P1 form | https://www.rcsb.org/structure/6DD9 | Protein Data Bank, 6DD9 |
| Rosenberg SC, Corbett KD | 2019 | X-Ray Diffraction data from M. musculus SYCP3 residues 105-248, source of 6DD8 structure | https://data.sbgrid.org/dataset/583/ | SBGrid Data Bank, 583 |
| Rosenberg SC, Corbett KD | 2019 | X-Ray Diffraction data from M. musculus SYCP3 residues 105-248, source of 6DD9 structure | https://data.sbgrid.org/dataset/584/ | SBGrid Data Bank, 584 |

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
