## [Decision Letter]

Thank you for submitting your article "A conserved mechanism for meiotic chromosome organization through self-assembly of a filamentous chromosome axis core" for consideration by *eLife*. Your article has been reviewed by three peer reviewers, including Bernard de Massy as the Reviewing Editor and Reviewer #1, and the evaluation has been overseen by Jessica Tyler as the Senior Editor.

The reviewers have discussed the reviews with one another and the Reviewing Editor has drafted this decision to help you prepare a revised submission.

Summary

Meiotic chromosomes have a unique mode of organization with a differentiated axial structure which provides ways to anchor chromatin loops and which is extremely important for the control of all events taking place from prophase to metaphase and thus to ensure the proper segregation of chromosomes at the first meiotic division. Many components are known but the understanding of their molecular organization is still limited. The Corbett group has previously reported the important role and organization of Horma-domain proteins and specifically one motif called the closure motif that provides interaction to itself but also to other partners allowing to build multiprotein complexes as shown in *C. elegans*. This family of Horma-domain proteins is evolutionarily conserved. Using a range of biochemical and structural assays, proteomics and EM the authors show that the assembly mechanism of these axis core proteins and their interaction with other components of the meiotic chromosome axis is conserved in yeast, plants and mammals.

Overall the studies are well conducted and the data is clearly presented. Although some insights are speculative (see below), this work represents a potentially significant advance in understanding the role of axis core proteins in the formation and function of meiotic chromosome structure.

Essential revisions

Several experiments require clarification and further validations. In addition, some conclusions are highly speculative which is fine for a discussion but should not be presented as conclusions of this study per se and thus should not be included in the Abstract or title. Both title and Abstract should highlight the conclusions of the experiments and should be revised. It does not seem that the mechanism of meiotic chromosome organization is known yet: how filaments and bundles form or dissociate is unknown. How cohesins are build-in to contribute to loop formation is also unknown.

1) Red1 filaments and their role in vivo: the use and test of the chimeric protein (*Sc/Zr*) is important but a critical test that should be done is to mutate the I715 residue in the chimera.

Figure 1H: Please include *Sc* Red1 wild-type staining for comparison and quantify Red1 staining to support the statement that the construct "does not affect chromosome localization" (Figure 1H legend).

Does *Sc* Red1 I743A (or I743R) load onto chromosome axes as expected? The yeast imaging is only partially convincing. Protocols are available to actually see the Red1 axis by spreading nuclei which would be much more informative about the consequence of the mutations tested.

2) Clarify how the closure motif is defined and what experiment beyond the interaction data allows for the conclusion of the presence of a closure motif. In Figure 2A: show the sequence alignment of HORMAD-binding closure motifs of SYCP2, HORMAD1 and HORMAD2.

3) Figure 2—figure supplement 2A: data suggest that HORMAD2 FL interacts weakly with SYCP2 compared to the HORMA domain alone (residues 1-241). This suggests that the C-terminal closure motif of HORMAD2 can compete with the SYCP2 closure motif for HORMAD binding. Should test this using the in vitro pull-down assay shown in Figure 2—figure supplement 2C. The competition between closure motifs is likely an important feature of axis assembly/disassembly.

4) How filaments form is unclear. Which interactions are involved? Specific concerns are also raised by the model of Sycp2/3: using SAXS and XLMS the authors claim that the coiled-coil domains of Hs SYCP3 and SYCP2 form a heterotetramer with parallel SYCP3 molecules that are anti-parallel to (parallel) SYCP2 molecules (i.e. a parallel homo-dimer of anti-parallel heterodimers).

While the antiparallel orientation of SYCP3 with respect to SYCP2 is clearly supported by XLMS, this suggested organization results in a polar tetramer due to the polarized orientation of SYCP3 and SYCP2, respectively. Is there any evidence for such a polar axis core filament?

Can the authors exclude the possibility of an apolar heterotetramer with SYCP3 (and SCYP2) being antiparallel to both, SYCP2 and SYCP3 (i.e. an anti-parallel homo-dimer of anti-parallel heterodimers)?

Can the authors use their structural model to simulate XLMS data for more evidence in support of their model compared to alternatives?

5) Filament vs bundle. How general is the property to form bundle is unclear and how these bundle are formed is not clear: if the LQSMLF sequence only exists in mammalian SYCP3 (Figure 2H), please comment on how the bundling might occur in yeast and plants since bundling appears to be essential for chromosome axis formation?

6) Some citations in the Introduction seem arbitrary and the authors should verify that the references listed support their statements. For example, in paragraph four of the Introduction section the authors write "[…] lateral elements of the SC, become linked by coiled-coil transverse filaments along their entire length (Wojtasz et al., 2009, Smith and Roeder, 1997; Börner, Barot and Kleckner, 2008; Joshi et al., 2009, Roig et al., 2010 and Lambng et al., 2015)". The citations listed here refer to the role of Pch2, not the establishment/existence of transverse filaments.

Likewise, citations (Dunce et al., 2018, Lu et al., 2014, Cahoon et al., 2017, Köhler et al., 2017 and Schücker et al., 2015) do not only show the "molecular architecture of the SC transverse filaments and associated central element" but also include super-resolution studies of localizations of components within the lateral element, while a reference to the structure of central element proteins SYCE2-TEX12 (Davies et al., 2012) is missing.

---

## [Author Response]

Essential revisionsSeveral experiments require clarification and further validations. In addition, some conclusions are highly speculative which is fine for a discussion but should not be presented as conclusions of this study per se and thus should not be included in the Abstract or title. Both title and Abstract should highlight the conclusions of the experiments and should be revised. It does not seem that the mechanism of meiotic chromosome organization is known yet: how filaments and bundles form or dissociate is unknown. How cohesins are build-in to contribute to loop formation is also unknown.

We appreciate that the final sentence in our Abstract contained speculation about how our findings fit into the developing model of chromosome axis assembly. While we would argue that this sentence provided important context for the importance of the study and was not presented in a way that suggests we have demonstrated the claims, we have nonetheless removed this sentence from the Abstract. We hope that the revised Abstract is more satisfactory.

We have also re-written the title to focus more on the findings of the paper, as suggested.

*1) Red1 filaments and their role* in vivo*: the use and test of the chimeric protein (Sc/Zr) is important but a critical test that should be done is to mutate the I715 residue in the chimera.*

Figure 1H: Please include Sc Red1 wild-type staining for comparison and quantify Red1 staining to support the statement that the construct "does not affect chromosome localization" (Figure 1H legend).Does Sc Red1 I743A (or I743R) load onto chromosome axes as expected? The yeast imaging is only partially convincing. Protocols are available to actually see the Red1 axis by spreading nuclei which would be much more informative about the consequence of the mutations tested.

The reviewer is correct that the data presented in the original Figure 1H was not strong. The microscopy samples were in fact prepared using a popular spreading technique (Grubb… Bishop *JOVE* 2015), but the results were obviously less than ideal. To overcome these issues, we collaborated with Amy MacQueen (Wesleyan University) to perform high-quality imaging of wild-type *Sc* Red1 and three *Sc/Zr* chimeras: full-length (including residues 707-798 of *Zr* Red1), 707-791, and the 707-798(I715R) mutant as suggested. The results are presented in the new Figure 1H-I, and Figure 1—figure supplement 3. We imaged both Red1 and Gmc2, a component of the synaptonemal complex central element, in all strains. With these much-improved spreads, we observed that the full-length chimera localizes to chromosomes, albeit to a lesser extent than wild-type *Sc* Red1, and supports near-complete synaptonemal complex assembly. Both removal of residues 792-798 and mutation of I715 to R reduced (but did not eliminate) Red1 chromosome localization, and did not support synaptonemal complex assembly. This result is further discussed in the Results section.

Unfortunately, it is not known whether *Sc* Red1 I743A or I743R localizes to chromosomes. In the paper where this mutant was described (Eichinger et al., 2010), the authors showed that the *red1-I743A* strain has extremely low spore viability (5.7%, compared to 87.8% for wild-type *RED1*) and does not assemble synaptonemal complexes (as judged by Zip1 staining). This suggests that the I743A mutant may behave equivalently to our *Sc-Zr* chimera containing the I715R mutant. More is known about the Red1 I758R mutant (Lin et al., 2009), which is also located in the predicted coiled-coil region of Red1. This mutant shows low spore viability (<1%) and does not assemble synaptonemal complexes, but localizes strongly to meiotic chromosomes.

We compared the size-exclusion profiles of purified *Sc* Red1 (CTD) WT, I743R, and I758R, and found that both mutations strongly disrupt oligomer formation. Indeed I758R, being localized in the middle of the predicted coiled-coil region, appears to disrupt the large assemblies more effectively: I743R causes dissociation to a tetramer form, while I758R causes dissociation to individual monomers. These data were not included in the original manuscript, but are now included in Figure 1—figure supplement 1B, and discussed in subsection “Budding Red1 forms filaments from coiled-coil tetramer units” of the revised manuscript.

2) Clarify how the closure motif is defined and what experiment beyond the interaction data allows for the conclusion of the presence of a closure motif. In Figure 2A: show the sequence alignment of HORMAD-binding closure motifs of SYCP2, HORMAD1 and HORMAD2.

Figure 2A now shows a sequence alignment between the putative closure motifs of *M. musculus* SYCP2, HORMAD1, and HORMAD2. The new Figure 2—figure supplement 3 further shows a multiple-sequence alignment of SYCP2 and HORMAD1, with the motifs aligned as in Figure 2A.

Regarding the definition of a closure motif, we define it as a short sequence that binds to a HORMA domain protein in a manner similar to that of the known “MAD2-interacting motifs” of CDC20 and MAD1 (binding MAD2), or the closure motifs of the *C. elegans* meiotic HORMADs, which we have shown bind these proteins equivalently to MAD2-interacting motifs (Kim et al., 2014). We later showed using biochemical assays and HD-exchange that similar motifs in the *S. cerevisiae* Hop1 C-terminus and Red1 central region bind the Hop1 HORMA domain as closure motifs (West et al., 2018). We do not have direct data indicating that the putative closure motifs in mammalian and plant axis proteins are *bona fide* closure motifs. We are comfortable calling them closure motifs based on the following extremely strong circumstantial evidence:

1) HORMA domains are well-known to bind short peptide motifs in one and only one way.

2) We have identified short peptides that specifically bind the HORMA domain of meiotic HORMADs in both families.

3) These motifs form stable complexes with their respective HORMA domain partners in co-expression analysis (Figure 2—figure supplement 2C and Figure 5C).

4) The identified motifs are located similarly to the verified closure motifs in *S. cerevisiae*: at the extreme C-terminus of the HORMADs themselves, and immediately following an ordered N-terminal domain in the axis core proteins. As plant ASY3 lacks this ordered N-terminal domain, the identified closure motif is at the extreme N-terminus of this protein.

Finally, the biochemical data presented in the next response (and in the new Figure 2—figure supplement 2D) further supports that these are indeed closure motifs.

*3) Figure 2—figure supplement 2A: data suggest that HORMAD2 FL interacts weakly with SYCP2 compared to the HORMA domain alone (residues 1-241). This suggests that the C-terminal closure motif of HORMAD2 can compete with the SYCP2 closure motif for HORMAD binding. Should test this using the* in vitro *pull-down assay shown in Figure 2—figure supplement 2C. The competition between closure motifs is likely an important feature of axis assembly/disassembly.*

The reviewer is correct that the putative closure motif on the HORMAD2 C-terminus likely competes for binding the HORMA domain with the putative closure motif on SYCP2. We have previously shown that in *S. cerevisiae*, the Hop1 HORMA domain binds more strongly to the closure motif on Red1 than the closure motif on its own C-terminus (West et al., 2018). The reviewer is right to ask for similar analysis in the mammalian proteins. Unfortunately, the pulldown assay shown in Figure 2—figure supplement 2C was performed with the two proteins co-expressed in *E. coli*, then purified, making it non-ideal for the competition assay the reviewer suggests. Instead, we purified the HORMAD2 HORMA domain (residues 1-241) and tested its binding to a peptide encoding the HORMAD2 closure motif (residues 282-306), and measured a *Kd* of ~ 7 uM, in line with prior measurements for *C. elegans* and *S. cerevisiae* meiotic HORMADs binding their closure motifs in similar assays (Figure 2—figure supplement 2D). Importantly, we found that a HORMAD2^1-241^:SYCP2^390-429^ complex did not bind the same peptide, showing that the putative closure motifs from SYCP2 and HORMAD2 indeed compete for binding to the HORMAD2 HORMA domain.

4) How filaments form is unclear. Which interactions are involved?

We envision that filaments form from interdigitation of coiled-coil segments of Red1, SYCP2/SYCP3, or ASY3/ASY4 that extend past the core of an individual coiled-coil tetramer. This idea is supported by our finding that truncation of SYCP2, SYCP3, and Red1 all effectively eliminate filament formation but maintain tetramers.

Specific concerns are also raised by the model of Sycp2/3: using SAXS and XLMS the authors claim that the coiled-coil domains of Hs SYCP3 and SYCP2 form a heterotetramer with parallel SYCP3 molecules that are anti-parallel to (parallel) SYCP2 molecules (i.e. a parallel homo-dimer of anti-parallel heterodimers).While the antiparallel orientation of SYCP3 with respect to SYCP2 is clearly supported by XLMS, this suggested organization results in a polar tetramer due to the polarized orientation of SYCP3 and SYCP2, respectively. Is there any evidence for such a polar axis core filament?

The reviewer is correct that in our model, the SYCP2:SYCP3 and ASY3:ASY4 filaments (but not those of Red1) would be polar. At present, there is no additional data (in vitro or in vivo) supporting this idea.

Can the authors exclude the possibility of an apolar heterotetramer with SYCP3 (and SCYP2) being antiparallel to both, SYCP2 and SYCP3 (i.e. an anti-parallel homo-dimer of anti-parallel heterodimers)?Can the authors use their structural model to simulate XLMS data for more evidence in support of their model compared to alternatives?

We have favored a model in which the SYCP2:SYCP3 heterotetramer is built with two parallel SYCP2 monomers, arranged antiparallel to two SYCP3 monomers (“parallel antiparallel” model in Figure 4—figure supplement 3A) It is also possible that a 2:2 heterotetramer could form with a pair of antiparallel SYCP2 protomers, and a pair of antiparallel SYCP3 protomers (“antiparallel antiparallel” model in Figure 4—figure supplement 3A). We used two methods to distinguish between these models.

First, we systematically modeled parallel versus antiparallel arrangements of SYCP2-SYCP2, SYCP3-SYCP3, and SYCP2-SYCP3, identified those lysine pairs predicted to be within 20 or 30 Å of one another (and would therefore be expected to form crosslinks in XLMS), and compared these predictions to our XLMS data. This data is presented in the attached Microsoft Excel workbook listed as Table S4. While noisy and incomplete, the XLMS data generally support a parallel arrangement for SYCP2-SYCP2 interactions, a parallel arrangement for SYCP3-SYCP3 interactions, and an antiparallel arrangement for SYCP2-SYCP3 interactions. This data is most consistent with the “parallel antiparallel” model.

Second, we performed SAXS analysis on our fused SYCP3^CC^-SYCP2^CC^ construct, this time with an intact N-terminal maltose-binding protein tag (~43 kDa). The two MBP tags are located on the same end of the tetramer in the “parallel-antiparallel” model, and on opposite ends in the “antiparallel-antiparallel” model. Given the size of MBP, these two possibilities are easily distinguished by SAXS. The new data, shown in Figure 4—figure supplement 3 and discussed in subsection “SYCP2 is an interaction hub for the mammalian chromosome axis” of the revised manuscript, clearly indicates that the MBP tags are on the same end of the tetramer, further supporting the “parallel-antiparallel” model.

5) Filament vs bundle. How general is the property to form bundle is unclear and how these bundle are formed is not clear: if the LQSMLF sequence only exists in mammalian SYCP3 (Figure 2H), please comment on how the bundling might occur in yeast and plants since bundling appears to be essential for chromosome axis formation?

While we agree with the reviewer that the observed bundling of mammalian SYCP2:SYCP3 filaments raises several questions, we have not explored this behavior extensively enough to adequately answer these questions. In particular, the tendency to form bundles does not appear to be shared by budding-yeast or plant meiotic axis core protein filaments, suggesting that bundling, if indeed it does occur in cells, may be specific to mammals. We have added a note to this effect in subsection “SYCP2 is an interaction hub for the mammalian chromosome axis”.

6) Some citations in the Introduction seem arbitrary and the authors should verify that the references listed support their statements. For example, in paragraph four of the Introduction section the authors write "[…] lateral elements of the SC, become linked by coiled-coil transverse filaments along their entire length (Wojtasz et al., 2009, Smith and Roeder, 1997; Börner, Barot and Kleckner, 2008; Joshi et al., 2009, Roig et al., 2010 and Lambng et al., 2015)". The citations listed here refer to the role of Pch2, not the establishment/existence of transverse filaments.

We apologize for this error. We have changed the references cited for this sentence.

Likewise, citations (Dunce et al., 2018, Lu et al., 2014, Cahoon et al., 2017, Köhler et al., 2017 and Schücker et al., 2015) do not only show the "molecular architecture of the SC transverse filaments and associated central element" but also include super-resolution studies of localizations of components within the lateral element, while a reference to the structure of central element proteins SYCE2-TEX12 (Davies et al., 2012) is missing.

We apologize for leaving out this important reference. We have altered the wording of the sentence in question to better reflect our original intent, and also added the Davies et al. reference. We have also carefully checked all references in the manuscript to ensure that they are used properly.